
# Dust Induced Atmospheric Absorption Improves Tropical Precipitations In Climate Models

Yves Balkanski[1], Rémy Bonnet[2], Olivier Boucher[2], Ramiro Checa-Garcia[1] and Jérôme Servonnat[1]

[1]Laboratoire des Sciences du Climat et de l'Environnement, CEA-CNRS-UVSQ, IPSL, Gif-sur-Yvette, France

[2]Institut Pierre-Simon Laplace, Sorbonne Université-CNRS, Paris, France

*Correspondance to*: yves.balkanski@lsce.ipsl.fr

**Abstract.** The amount of shortwave radiation absorbed by dust has remained uncertain. We have developed a more accurate representation of dust absorption that is based on the observed dust mineralogical composition and accounts for very large particles. We analyze the results from two fully-coupled climate simulations of 100 years in terms of their simulated precipitation patterns against observations. A striking benefit of the new dust optical and physical properties is that tropical precipitations over Sahel, tropical North Atlantic and West Indian Ocean are significantly improved compared to observations, without degrading precipitations elsewhere. This alleviates a persistent bias in earth system models that exhibit a summer African monsoon that does not reach far enough North. We show that the improvement results from a thermodynamical and dynamical response to dust absorption is unrelated to natural variability. Aerosol absorption induces more water vapor advection from the ocean to the Sahel, thereby providing an added supply of moisture available for precipitation. This work thus provides a path towards improving precipitation patterns in these regions by more realistically accounting for both physical and optical properties of the aerosol.

## 1. Introduction

Mineral dust influences precipitation through direct radiative forcing (Miller et al., 2014), changing the vertical temperature profile, and is an efficient ice nucleus in the presence of feldspar mineral (Atkinson et al., 2013), therefore also producing an indirect, cloud-mediated radiative perturbation. It also influences the water cycle through microphysical interactions with clouds (Nenes et al., 2014). Near source regions, mineral dust absorption causes a change in atmospheric radiation of several tens of watts per square meter ($W.m^{-2}$), an effect



stronger than the one exerted by aerosol-cloud interactions (Miller et al., 2014; Nenes et al., 2014). Sahel precipitation is influenced by aerosol absorption (Miller et al., 2004; Solmon et al., 2008; Yoshioka et al., 2007), and absorption depends on iron oxides (hematite and goethite) that are part of dust mineralogical composition

(Claquin et al., 1999; Sokolik and Toon, 1996). Over the last 15 years, simulating tropical precipitation has been notoriously difficult for climate models (Fiedler et al., 2020). Improving the representation of tropical monsoons is a prerequisite to predict future changes in tropical precipitations and attribute them to observed changes in greenhouse gases and to aerosol changes. We show here how a better representation of dust aerosols leads to an unequivocal improvement in the simulation of precipitation over key climatic tropical regions, namely Sahel,

tropical North Atlantic and West Indian Ocean without degrading precipitation elsewhere around the globe, and subsequently discuss the thermodynamically and dynamically-driven mechanisms at play that affect the water cycle.

Miller et al. (2014) showed that the increase in Sahel precipitation in response to high dust absorption is a fairly robust result across models. The link between this additional atmospheric absorption and dust physical

properties, however, remains poorly understood. Conversely, (Haywood et al., 2016) discuss how some tropical precipitation biases can be reduced by changing the model's energy balance between the Northern and the Southern Hemispheres, but they did so through ad hoc hemispheric albedo changes. Here we reunite these incomplete studies by describing an end-to-end physical mechanism that ties improvement in tropical precipitation to observational support for a higher level of dust absorption based on measurements of iron oxide

in dust particles, measurements of the full dust particle size distribution and detailed climate simulations with interactive dust.

The abundancy and variation in iron oxides, as well as the presence of large particles control absorption (Balkanski et al., 2007; Miller et al., 2004; Ryder et al., 2018; Yoshioka et al., 2007). Although iron oxides in dust are present in minute quantities from 1 to 5% by volume (Di Biagio et al., 2019; Journet et al., 2014;

Kandler et al., 2007; Liu et al., 2018; Nickovic et al., 2012; Perlwitz et al., 2015; Reid, 2003), this relatively small volume largely controls mineral dust absorption (Balkanski et al., 2007; Di Biagio et al., 2019; Ryder et al., 2018). Large dust particles have been shown through recent measurements to absorb strongly (Ryder et al., 2018). Since large particles are particularly abundant over source regions, they change their atmospheric energy balance. Furthermore Sahel is one of the world regions with the highest iron oxide content in soils (Journet et

al., 2014; Nickovic et al., 2012). In this study, we analyse the strong relationship between high iron oxide content and increased Sahel precipitation. Specifically we determine the optical properties of airborne dust coming from African deserts based on its mineralogy and show the IPSL-CM6 precipitation fields in several key





tropical areas are improved compared to observations when the effect of dust absorption is introduced in this model. We finally dissect the mechanisms that explain an increase in Sahel precipitation with dust absorption

and  answer the question of whether these mechanisms are thermodynamical, dynamical or occur in response to an improved phasing of the Atlantic Multidecadal Variability.

## 2.  **Results**

We present in this study results from a fully-coupled climate simulation of 100 years (from which we analyse the last 30) and compare simulated precipitations with observations. Iron oxide content is based on the observations

of dust mineralogical composition over the Sahel region (Di Biagio et al., 2019; Lafon et al., 2006) and displayed for soils in Fig. 1. We infer the refractive index of the mineral dust using an optical model. Figure 2 illustrates the influence of the iron oxide content and the size of a particle on its radiative absorption. In this Figure, the aerosol absorption increases with the aerosol co-single-scattering-albedo (coSSA) along the x-axis. The coSSA is defined as:

$$coSSA = 1.-SSA \qquad\qquad\qquad\qquad\qquad\qquad\qquad\qquad\qquad\qquad (1)$$

the solid blue line illustrates the absorption calculated when only particles of dust of less than 10 μm are considered. Furthermore, the shift from the solid blue to the solid orange line indicates dust absorption increase when particles larger than 10 μm are taken into account. Hence, looking at the graph, particles of diameter less than 10 μm with an iron oxide content of 5.0% absorb the same amount of radiation than particles with 3.0%

iron oxide for which we consider also the diameters greater than 10 μm. With these mineralogical compositions we obtain a coSSA of 0.09. Considering dust particles with iron oxide volume content of 1.5%, thought to be the global median value as discussed in Balkanski et al. (2007), the coSSA is 0.032 and it almost doubles to 0.058 when large particles are also considered. Hence, large particles, which are particularly abundant over the Sahel region, increase substantially the aerosol absorption. This has yet to be taken into account in many models since

they do not generally include particles sizes above 10 μm.

We now discuss the changes in aerosol direct radiative effect for dust containing 3.0% iron oxide which corresponds to the amount of iron oxides measured over Sahel (Fig. 1).  Figure 3 illustrates the June, July, August and September (JJAS) mean radiative perturbation due to the presence of dust. Over the bright surfaces of the Sahel region, the top-of-atmosphere shortwave radiative perturbation is positive, i.e., an atmosphere with

dust is less reflective than an atmosphere without dust. At the surface, the radiative effect from dust is strongly negative (-18 W.m$^{-2}$ over the Sahel) as the dominant term is the reduction of shortwave radiation due to either



back-scatter or absorption of radiation by dust. The difference between TOA and surface determines the dust JJAS mean atmospheric absorption due to dust (+26 W m$^{-2}$ over the Sahel). Since dust is highly variable in time, particularly strong dust episodes are characterized by atmospheric absorption that reaches several hundred watts per square meter (Pérez et al., 2006). Note that, in comparison, greenhouse gases contribute to a globally-averaged radiative forcing of only 3 W m$^{-2}$ (Myhre et al., 2013) relatively constant on short timescales .

To evaluate the impact of this change of radiative forcing on precipitation, we calculate the difference in 30-year (1985-2014) precipitation averaged over June, July, August and September (JJAS) between the simulation with dust and the one with no dust (Figs 4 & 5). These figures show an increase in precipitation between 6 and 20°N latitude over Africa. A general feature of most ESMs is to have a summer African monsoon that does not reach far enough North compared to observations such as Tropical Rainfall Measuring Mission (Roehrig et al., 2013) (TRMM). Figure 5 shows that precipitation averaged from 10°W to 10°E increases from 0.5 to 1.5 mm day$^{-1}$ over the summer months (JJAS). Accounting for dust absorption hence shifts northward the extent of the African summer monsoon.

We now examine whether precipitations are better represented when the effect absorbing dust on atmospheric heating is accounted for. The accuracy with which the model captures the African monsoon was analyzed by comparing the incursion of the precipitations into the African continent with the TRMM observations for the same period of 15-years (2000-2014) over the summer months (JJAS). Figure 6 introduces the change in water budget due to the effect of absorbing dust over an airshed of the size of the Sahel (10°N-20°N; 15°W- 35°E). We derive the amount of water advected into this region with the same units than precipitation or evaporation (mm day$^{-1}$, see Methods). The effect of dust on precipitation is to increase the advection of moisture entering Sahel from the southern edge at 10°N. Figure S1 shows that aerosol absorption induces more water vapor advection from the ocean to the Sahel, thereby providing an added supply of moisture that is available for precipitation. When sufficient water vapor is advected towards the African continent, aerosol absorption changes the regional energy balance and enhances vertical exchanges through the air column (Miller et al., 2014; Solmon et al., 2008). The largest change in water flux into the Sahel airshed is through the southern border of the region at 10°N with a flux reduction of 89% due to dust absorption, from -0.41 mm day$^{-1}$ (i.e. exiting the Sahel box) to only -0.05 mm day$^{-1}$. The change in flux on the western side of the airshed is significantly smaller and amounts to 0.1 mm day$^{-1}$. By comparison, the increases in precipitation and evaporation over the Sahel region are 0.40 and 0.09 mm day$^{-1}$, respectively, for the months JJAS averaged over the 30-year period (1985-2014).



## 3. Discussion

We now compare the distribution of the surface precipitation between the two model simulations with and without dust with observations from the Global Precipitation Climatology Project (GPCP) for the months when African monsoon sets in from June to September. The precipitation statistics based upon the model/measurements comparison show significant improvements over Sahel, North Africa and the North Atlantic. We quantify these improvements in Table 1 through a comparison of the bias, the root mean square error (RMSE) and the spatial correlation between precipitation fields. The largest changes occur over the Sahel where precipitation increases by 21% over the period and the negative bias and of the RMSE are reduced by 34 and 29%, respectively. The spatial correlation of the precipitation is also improved from 0.951 to 0.965. Other regions where all of the mentioned statistics are improved are the North Atlantic and the West Indian Ocean. Over Northern Africa, the strong dust absorption causes an excess precipitation but there are improvements in the RMSE and the spatial correlation.

The Sahel region is prone to substantial atmospheric dust absorption as it is an important mineral dust source region. Being an active source region, the lower troposphere above Sahel experiences high dust loads and a large mass fraction comprises large-size particles of diameters above 10 μm (larger particles being more absorbing than smaller ones). Compounding these effects, soils from Sahel have a higher iron oxide content than other soils from North Africa (Di Biagio et al., 2019; Formenti et al., 2014; Kandler et al., 2007; Liu et al., 2018). The high atmospheric absorption that exists over the Sahel and illustrated by Fig. 3, induces water moisture advection from the North Atlantic in regions off the West Coast of Africa during the period of the African summer monsoon (Fig. S1). We present below the terms of the energy budget that play a role over the Sahel have been discussed by Miller et al. (2014). We also explain the main elements that lead to upward movement of the moist air above the Sahel region that then generates precipitation. The action of mineral dust aerosol on precipitation is felt through the modifications of the diabatic heating and how the aerosol affects evaporation.

The terms in the energy balance that influence evaporation are the dimming caused by the aerosol layer of -18 W.m$^{-2}$ over the Sahel (see Fig. 3) which needs to be compensated by the decrease of net latent heat flux that leaves the surface. Following Miller et al.(2014), this surface flux reduction can be expressed as:

$$F_{surf} = \partial R_{surf}^{LW} + \partial LE + \partial S_E \tag{2}$$

where $\partial R_{surf}^{LW}$ represents the change upward LW radiation flux, and the two other terms, $\partial LE$ and $\partial S_E$ are the changes in turbulent fluxes of latent and sensible heat due to the presence of dust. Over the oceans, the change in



latent heat flux dominates and the evaporation is reduced hence diminishing precipitation over these regions (see last column of Table 1). Figure S2 indicates the vertical amount of transported Moist Static Energy (MSE). Two regions can be distinguished, one over the Sahel where the transport is upward as a result of the absorbed energy by dust, and the surrounding regions with a downward motion. These are the main perturbations that accompany the precipitation anomaly caused by the presence of dust. Other variables that increase are: evaporation, low-

level cloud cover, liquid water path. As noticed by Miller et al. (2004), and reinforced by the results presented here, evaporation over Sahel increases with dust absorption (see Fig. S3), while precipitation increases over the same region. Part of the evaporation is supplied by the moisture brought through precipitation over the same region. One mechanism for supplying the moisture is the advection that is evidenced in Fig. S1 and in the water budget presented on Fig. 6.

To complete the analysis of the possible mechanisms that explain this improvement in Sahel precipitation, we examined whether a different phasing of the Atlantic Multidecadal Variability (Enfield et al., 2001) (AMV), a basin-wide low-frequency variations of the sea surface temperature over the North Atlantic, could be responsible for better reproducing observed precipitation fields. Figure S4 compares the observed and simulated AMV Index (Enfield et al., 2001; Trenberth and Shea, 2006). The AMV index is more in phase in the simulation without dust

than in the simulation with dust. We conclude from this analysis that the precipitation improvements brought about by dust absorption, which are also evident for earlier portions of the 100-years long simulations, are not due to a better phasing of natural variability in the dust simulation. Instead they correspond to dynamical effects that respond to a thermodynamically driven forcing.

## 4. Conclusion

This study was designed to realistically represent dust absorption over the Sahel region and describe the mechanisms by which dust stimulates summer precipitation (JJAS) over the region. Our modelling includes two aspects not accounted for prior to it, first accounting for the high iron oxide content of the region in the optical properties, and secondly taking into consideration the extra-absorption from very large particles, with diameters greater than 10 μm, generally not represented in climate models. The striking benefit to estimate more precisely

dust absorption and take into account very large particles is that, at least in the IPSL-CM6 model[1] considered here, tropical precipitations are significantly improved compared to observations. This important result came

---

[1] IPSL-CM6 is the Earth System Model developed at the Institut Pierre- Simon Laplace and described in Boucher et al.(Boucher et al., 2020)





almost serendipitously, as we set to check if the comparison between simulated and observed precipitation over 30 years showed improvement. In key regions of tropical precipitations, namely: Sahel, tropical North Atlantic and West Indian Ocean the precipitation in the IPSL-CM6 climate model are significantly improved without

degrading precipitations elsewhere. We believe that this is not restricted to this climate model as other models participating in the CMIP exercises have the same bias over Sahel, which is to have to little advection of water vapor northward into the region during the Northern Hemisphere summer months. We thus provide a path towards improving precipitation patterns in these regions by more realistically accounting for both physical (size-based) and optical (absorption) properties of the aerosol. Our results also offer a strong physical basis for a

stabilizing feedback loop involving dust emission, atmospheric absorption, Sahel precipitation, and vegetation, as hypothesized by Carslaw et al. (2013), which could create multiannual or multidecadal oscillations at these latitudes that could also interact with natural models of variability in the Atlantic region. Future studies should therefore account for the role of water recycling from semi-arid vegetation (Yu et al., 2017) which plays a potentially important role in this loop.






## 5. Appendix

### 5.1 IPSL-CM6 description

The climate model used here is the low resolution model from the Institut Pierre-Simon Laplace Climate
Modelling Centre described by Boucher et al. (2020). The horizontal resolution is 2.5° in longitude and 1.28° in
latitude with a discretization of the vertical into 79 layers that extends to about 80 km. For the ocean model,
NEMO, that includes sea-ice and biogeochemistry, the horizontal resolution is 1° and the model is discretized
using 75 vertical levels. The aerosols are run interactively in the simulations presented here.

### 5.2 Dust modeling

Dust emission fluxes are calculated in two steps: in a first step, we derive the horizontal flux of dust that is
mobilized based upon three criteria: a threshold velocity that depends on the nature of the upper soil, the wind
speed at 10-meters and an erodibility factor that takes into account the effect of soil moisture. These erodibility
factors were tuned following the procedure described in Balkanski et al. (2004). Total emissions and loads
compare well with the constraints given by Ridley et al. (2016) and by Kok et al. (2017). The dust particle size
distribution are emitted with a constant shape following the Brittle theory described by Kok (2011). The size
distribution is represented by one or several modes represented by a log-normal distributions with a mass median
diameter which varies in response to the sink processes of the dust cycle. Simulations are done either with one
mode centered at 2.5 µm with a width of 2.0 that represents the accumulation and coarse mode (Denjean et al.,
2016; Schulz et al., 1998). Accounting for large particles of more than 10.0 µm follows a treatment of the size
distribution with four modes (Di Biagio et al., 2020). The four-mode distribution has mass median diameters of
1.0, 2.5, 7.0 and 22.0 µm, respectively. The mineral composition which is described below is chosen to have the
same dust absorption on all simulations.

Dust absorption is influenced mainly by the iron oxide embedded in the dust aggregates (Di Biagio et al., 2019;
Lafon et al., 2006; Ryder et al., 2018). Measurements of iron oxides on soils from around the world have been
reported for two particle size class the clays with diameters of less than 2 microns and the silts with diameters
between 2 and 64 µm. The iron oxide varies drastically depending on the soil types and most measurements
indicate a weight content of 1 to 7% equivalent to 0.5 to 3.5% (Di Biagio et al., 2019; Engelbrecht et al., 2016;
Journet et al., 2014; Lafon et al., 2006; Moosmüller et al., 2012)  by volume. To determine the amount of iron
oxides over Sahel, we used the high-resolution database published by Nickovic et al. (2012) and determined the
amount of hematite over the Sahel region (16°W-36°E; 10°N-20°N), see Fig. 1. With the 30 s grid high-





resolution of the database, the mineral content of hematite could be retrieved for 6,026,016 points. For the clay

fraction (diameter ≤ 2 μm), 50% of these points had and hematite content of more than 2% by weight (equivalent

to 1% by volume since density of hematite is twice that of all other minerals except goethite); 30% (respectively

17%) of the points had an hematite content of more than 3% (resp. 4%) by weight. For the silt fraction (diameter

> 2 μm), 49% of these points had and goethite content of more than 2% by weight (equivalent to 1% by volume

since density of hematite is twice that of all other minerals except goethite); 30% (respectively 12%) of the

points had an hematite content of more than 4% (resp. 5%) by weight. Assuming that hematite and goethite

contents are the same for these soils and accounting for the density of hematite (5300 kg m$^{-3}$) and goethite (3800

kg m$^{-3}$), we estimate that iron-oxides (hematite+goethite) represent 5.3% by weight and 3.0% by volume of

mineral dust that has a density of 2650 kg m$^{-3}$. Hence, in the simulation, we took the optical properties of

mineral dust with a volume of  3.0% made of iron oxides.

The absorption of dust size distribution is determined as follows: we consider dust as the mixture of six minerals

kaolinite (kaol), illite (lili), montmorillonite (montmo), quartz (qua), calcite (calci) and hematite (hema). The

difference in optical properties between goethite and hematite are not considered in this paper as we focus on the

mechanisms by which dust absorption causes an increase of precipitation over the Sahel and not into having a

very precise calculation of this absorption. We vary the VOLUME content of hematite with the following

values: 0.9, 1.5, 2.7, 3.0, 4.0, 5.0 and 10%.

A description of the Maxwell-Bruggeman approximation used here to compute the refractive index of dust can

be found in Balkanski et al. (2007). The first step of the computation is for each of these hematite content

compute the refractive index of the mixtures: kaol-hema, illi-hema, montmo-illi, qua-hema and calci-hema  (for

example for 3% hematite, all mixtures are composed of 97-3%). The second step is to compute the refractive

index for the two most abundant constituents of the mixture as clays associated with hematite that is illi-hema

and kaol-hema. The third step takes the resulting mixture illi-kaol-hema and mixes it with the third most

abundant mineral montmorillonite. The resulting mixture illi-kaol-montmo-hema is then mixed with the fourth

most abundant mineral quartz. And finally the mixture illi-kaol-montmo-quartz is mixed with calcite, the least

abundant of those minerals. We refer the reader to Table 1 that explains the abundancies of the different

assemblages and minerals.

Figure 2 illustrates how co-albedo (1.-SSA),  SSA (single scattering albedo) varies with increasing iron oxide

content and the effect of considering large particles (diameter > 10 μm). For a co-albedo of 0.09, we can see

from these curves that the same absorption from the whole size distribution including large particles (orange

solid line) requires only 3.0% volume content of iron oxide whereas for particles of less than 10 μm in diameter





the have to include a volume of 5.0% of iron oxide. This is also observed in field measurements (see Fig. 8 from Ryder et al. ( 2013)).

We ran 100-year simulation of the fully-coupled IPSLCM6 model with all interactive components of the aerosol including dust for the 1915-2014 period, as well as another 100-year simulation without the dust. We analyzed

the last 30 years of the coupled simulations (1985-2014) for the summer period that includes the month of June (June-July-August-September) referred to as JJAS in the rest of the text. We checked for all variables the consistency of the results compared to the previous 30-year period from 1955 to 1984.

### 5.3   **Computation of the Direct Radiative Perturbation (DRP) from dust**

We compare the fluxes both at top-of-atmosphere and at the surface for the simulation with the dust aerosol and

the case when dust aerosol concentrations are set to zero in the model. The refractive indices are described for the shortwave in the work from Di Biagio et al. (2019) and Balkanski et al. (2007) and for the longwave from Di Biagio et al. (2020). The solar radiation code in the LMDZ GCM consists of an improved version of the parameterizations of Fouquart and Bonnel (1980). The radiative transfer module includes a six‑band (0.185–4.0 µm) scheme in the SW and the Rapid Radiative Transfer Model for Global Circulation Models radiative scheme

in sixteen bands between 3.33 and 1,000 µm (Hogan and Bozzo, 2016). The model accounts for the diurnal cycle of solar radiation and allows fractional cloudiness to form in a grid box. The reflectivity and transmissivity of a layer are computed using the delta Eddington approximation(Joseph et al., 1976) in the case of a maximum random overlap(Morcrette and Fouquart, 1986) by averaging the clear and cloudy sky fluxes weighted linearly by their respective fractions in the layer. The radiative fluxes are computed every two hours, at the top-of-

atmosphere and at the surface, with and without the presence of clouds, and with and without the presence of aerosols. The clear-sky and all-sky aerosol radiative forcings can then be estimated as the differences in radiative fluxes with and without aerosols.

Few models have published the radiative effect of dust over the Sahel, hence to compare this effect to other publications we present in Table S1 global results and discuss how the net atmospheric absorption compares in

each model. Most models published predict a net absorption of radiation by dust except for the result from Yoshioka et al. (Yoshioka et al., 2007) that we discuss below. If we take the three following studies Woodward (2001) and Miller et al. (2004, 2014) they have in common that the absorption in the shortwave dominates what is absorbed in the longwave. Compared to this study, where LW absorption represents -0.41 W.m$^{-2}$, the LW absorption of these 3 papers are within the range [-0.23 to -0.03 W.m$^{-2}$], i.e., the net surface LW radiation term is

greater than the one at top-of-atmosphere.





In the study of Yoshioka et al. (2007) the top-of-atmosphere LW radiation effect is within the other models but the surface LW effect amounts to +1.13 W.m$^{-2}$, nearly twice the effect of this study. The consequence is that the atmospheric cooling caused by the LW effect (-0.81 W.m$^{-2}$) more than compensates for the SW heating (+0.67 W.m$^{-2}$) and results in a net cooling of the atmospheric column.

### 5.4 **Deriving the Water Budget along the Sahel airshed (10°N-20°N; 15°W- 35°E)**

Following Sheen et al.(Sheen et al., 2017), we seek to determine the total flux of moisture depth across each of the airshed boundaries. Hence we compute the integrated moisture flux through the following integral:

$$1./\rho_w g \int_0^{ps} \langle qu \rangle dp \tag{1}$$

where $\langle qu \rangle$ represents the monthly mean of the $qu$ product, $g$ is the acceleration due to gravity, $\square_w$ is the density

of water and $p_s$ is the surface pressure. The units for the flux are kg.m$^{-1}$ s$^{-1}$. The integrated fluxes across each side of the airshed are first averaged and then scaled by the length along the flux trajectory(Trenberth, 1999), and then divided by the airshed area to obtain units of mm per day that can be compared to the precipitation and the evaporation fluxes over the airshed.

*Deriving the vertical advection response of moist static energy over North Africa (Fig. S2)*

Following the method of Hill et al.(Hill et al., 2017), the following term allows to estimate the vertical advection of Moist Static Energy (MSE):

$$\delta \left( \langle \omega \rangle \frac{\partial \langle MSE \rangle}{\delta p} \right) \tag{2}$$

Where MSE, the moist static energy is derived as:

$$MSE = c_p T + gz + L_V \, ovap \tag{3}$$

$c_p$: is the heat capacity of dry air

$T$ is the temperature

$gz$ is the geopotential height

$L_v$ is the latent heat of vaporization of water

*ovap*: specific humidity






■ **References**

Atkinson, J. D., Murray, B. J., Woodhouse, M. T., Whale, T. F., Baustian, K. J., Carslaw, K. S., Dobbie, S., O'Sullivan, D. and Malkin, T. L.: The importance of feldspar for ice nucleation by mineral dust in mixed-phase clouds, Nature, 498, 355, 2013.

Balkanski, Y., Schulz, M., Moulin, C. and Ginoux, P.: : The formulation of dust emissions on global scale: formulation and validation using satellite retrievals, in Emissions of Atmospheric Trace Compounds, eds. C. Granier, P. Artaxo and C. Reeves, pp. 239–267, Kluwer Academic Publishers, Dordrecht, , 2004.

Balkanski, Y., Schulz, M., Claquin, T. and Guibert, S.: Reevaluation of Mineral aerosol radiative forcings suggests a better agreement with satellite and AERONET data, Atmospheric Chem. Phys., 7(1), 81–95, 310    https://doi.org/10.5194/acp-7-81-2007, 2007.

Boucher, O., Servonnat, J., Albright, A. L., Aumont, O., Balkanski, Y., Bastrikov, V., Bekki, S., Bonnet, R., Bony, S., Bopp, L., Braconnot, P., Brockmann, P., Cadule, P., Caubel, A., Cheruy, F., Codron, F., Cozic, A., Cugnet, D., D'Andrea, F., Davini, P., Lavergne, C., Denvil, S., Deshayes, J., Devilliers, M., Ducharne, A., Dufresne, J., Dupont, E., Éthé, C., Fairhead, L., Falletti, L., Flavoni, S., Foujols, M., Gardoll, S., Gastineau, G., 315    Ghattas, J., Grandpeix, J., Guenet, B., Guez, L., E., Guilyardi, E., Guimberteau, M., Hauglustaine, D., Hourdin, F., Idelkadi, A., Joussaume, S., Kageyama, M., Khodri, M., Krinner, G., Lebas, N., Levavasseur, G., Lévy, C., Li, L., Lott, F., Lurton, T., Luyssaert, S., Madec, G., Madeleine, J., Maignan, F., Marchand, M., Marti, O., Mellul, L., Meurdesoif, Y., Mignot, J., Musat, I., Ottlé, C., Peylin, P., Planton, Y., Polcher, J., Rio, C., Rochetin, N., Rousset, C., Sepulchre, P., Sima, A., Swingedouw, D., Thiéblemont, R., Traore, A. K., Vancoppenolle, M., 320    Vial, J., Vialard, J., Viovy, N. and Vuichard, N.: Presentation and Evaluation of the IPSL-CM6A-LR Climate Model, J. Adv. Model. Earth Syst., 12(7), https://doi.org/10.1029/2019MS002010, 2020.

Carslaw, K. S., Lee, L. A., Reddington, C. L., Pringle, K. J., Rap, A., Forster, P. M., Mann, G. W., Spracklen, D. V., Woodhouse, M. T., Regayre, L. A. and Pierce, J. R.: Large contribution of natural aerosols to uncertainty in indirect forcing, Nature, 503, 67, 2013.

Claquin, T., Schulz, M. and Balkanski, Y. J.: Modeling the mineralogy of atmospheric dust sources, J. Geophys. Res. Atmospheres, 104(D18), 22243–22256, https://doi.org/10.1029/1999JD900416, 1999.

Denjean, C., Cassola, F. and Mazzino, A., Triquet, S., Chevaillier, S., Grand, N., Bourrianne, T., Momboisse, G., Sellegri, K., Schwarzenbock, A., Freney, E., Mallet, M., and Formenti, P.: Size distribution and optical properties of mineral dust aerosols transported in the western Mediterranean, Atmos Chem Phys, (16), 1081– 330    1104, https://doi.org/10.5194/acp-16-1081-2016, 2016.

Di Biagio, C., Formenti, P., Balkanski, Y., Caponi, L., Cazaunau, M., Pangui, E., Journet, E., Nowak, S., Andreae, M. O., Kandler, K., Saeed, T., Piketh, S., Seibert, D., Williams, E. and Doussin, J.-F.: Complex refractive indices and single scattering albedo of global dustaerosols in the shortwave spectrum and relationship to iron content and size, Atmospheric Chem. Phys., 19, https://doi.org/10.5194/acp-2019-145, 2019.



Di Biagio, C., Balkanski, Y., Albani, S., Boucher, O. and Formenti, P.: Direct Radiative Effect by Mineral Dust Aerosols Constrained by New Microphysical and Spectral Optical Data, Geophys. Res. Lett., 47(2), https://doi.org/10.1029/2019GL086186, 2020.

Enfield, D. B., Mestas-Nuñez, A. M. and Trimble, P. J.: The Atlantic Multidecadal Oscillation and its relation to rainfall and river flows in the continental U.S., Geophys. Res. Lett., 28(10), 2077–2080,
https://doi.org/10.1029/2000GL012745, 2001.

Engelbrecht, J. P., Moosmüller, H. and Pincock, S., Jayanty, R. K. M., Lersch, T., and Casuccio, G.: Technical note: Mineralogical, chemical, morphological, and optical interrelationships of mineral dust re-suspensions, Atmos Chem Phys, 16 https://doi.org/10.5194/acp-16-10809-2016, 2016.

Fiedler, S., Crueger, T., D'Agostino, R., Peters, K., Becker, T., Leutwyler, D., Paccini, L., Burdanowitz, J.,
Buehler, S. A., Cortes, A. U., Dauhut, T., Dommenget, D., Fraedrich, K., Jungandreas, L., Maher, N., Naumann, A. K., Rugenstein, M., Sakradzija, M., Schmidt, H., Sielmann, F., Stephan, C., Timmreck, C., Zhu, X. and Stevens, B.: Simulated Tropical Precipitation Assessed across Three Major Phases of the Coupled Model Intercomparison Project (CMIP), Mon. Weather Rev., 148(9), 3653–3680, https://doi.org/10.1175/MWR-D-19-0404.1, 2020.

Formenti, P., Caquineau, S., Chevaillier, S., Klaver, A., Desboeufs, K., Rajot, J. L., Belin, S. and Briois, V.: Dominance of goethite over hematite in iron oxides of mineral dust from Western Africa: Quantitative partitioning by X-ray absorption spectroscopy, J. Geophys. Res. Atmospheres, 119(22), 12,740-12,754, https://doi.org/10.1002/2014JD021668, 2014.

Haywood, J. M., Jones, A., Dunstone, N., Milton, S., Vellinga, M., Bodas-Salcedo, A., Hawcroft, M., Kravitz,
B., Cole, J., Watanabe, S. and Stephens, G.: The impact of equilibrating hemispheric albedos on tropical performance in the HadGEM2-ES coupled climate model, Geophys. Res. Lett., 43(1), 395–403, https://doi.org/10.1002/2015GL066903, 2016.

Hill, S. A., Ming, Y., Held, I. M. and Zhao, M.: A Moist Static Energy Budget–Based Analysis of the Sahel Rainfall Response to Uniform Oceanic Warming, J. Clim., 30(15), 5637–5660, https://doi.org/10.1175/JCLI-D-
360 16-0785.1, 2017.

Hogan, R. and Bozzo, A.: ECRAD: A new radiation scheme for the IFS, ECMWF., 2016.

Joseph, J. H., Wiscombe, W. J. and Weinman, J. A.: The delta-Eddington approximation for radiative flux transfer, J Atmos Sci, 33, 2452–2459, 1976.

Journet, E., Balkanski, Y. and Harrison, S. P.: A new data set of soil mineralogy for dust-cycle modeling,
Atmospheric Chem. Phys., 14(8), 3801–3816, https://doi.org/10.5194/acp-14-3801-2014, 2014.

Kandler, K., Benker, N., Bundke, U., Cuevas, E., Ebert, M., Knippertz, P., Rodríguez, S., Schütz, L. and Weinbruch, S.: Chemical composition and complex refractive index of Saharan Mineral Dust at Izaña, Tenerife (Spain) derived by electron microscopy, Atmos. Environ., 41(37), 8058–8074, https://doi.org/10.1016/j.atmosenv.2007.06.047, 2007.



Kok, J. F.: Does the size distribution of mineral dust aerosols depend on the wind speed at emission?, Atmos Chem Phys, 11, 10149–10156, https://doi.org/10.5194/acp-11-10149-2011, 2011.

Kok, J. F., Ridley, D. A., Zhou, Q., Miller, R. L., Zhao, C., Heald, C. L., Ward, D. S., Albani, S. and Haustein, K.: Smaller desert dust cooling effect estimated from analysis of dust size and abundance, Nat. Geosci., 10(4), 274–278, https://doi.org/10.1038/ngeo2912, 2017.

Lafon, S., Sokolik, I. N., Dajot, J. L., Caquineau, S. and Gaudichet, A.: Characterization of iron oxides in mineral dust aerosols: Implications for light absorption, J Geophys Res Atm, 111(D21207), https://doi.org/10.1029/2005JD007016, 2006.

Liu, D., Taylor, J. W., Crosier, J., Marsden, N., Bower, K. N., Lloyd, G., Ryder, C. L., Brooke, J. K., Cotton, R., Marenco, F., Blyth, A., Cui, Z., Estelles, V., Gallagher, M., Coe, H. and Choularton, T. W.: Aircraft and ground
measurements of dust aerosols over the west African coast in summer 2015 during ICE-D and AER-D, Atmospheric Chem. Phys., 18(5), 3817–3838, https://doi.org/10.5194/acp-18-3817-2018, 2018.

Miller, R. L., Tegen, I. and Perlwitz, J.: Surface radiative forcing by soil dust aerosols and the hydrologic cycle, J. Geophys. Res. Atmospheres, 109(D04203), 4085–4112, https://doi.org/10.1029/2003JD004085, 2004a.

Miller, R. L., Tegen, I. and Perlwitz, J.: Surface radiative forcing by soil dust aerosols and the hydrologic cycle,
J. Geophys. Res. Atmospheres, 109(D4), https://doi.org/10.1029/2003JD004085, 2004b.

Miller, R. L., Knippertz, P., Pérez García-Pando, C., Perlwitz, J. P. and Tegen, I.: Impact of Dust Radiative Forcing upon Climate, in Mineral Dust: A Key Player in the Earth System, edited by P. Knippertz and J.-B. W. Stuut, pp. 327–357, Springer Netherlands, Dordrecht, https://doi.org/10.1007/978-94-017-8978-3_13, , 2014a.

Miller, R. L., Knippertz, P., Pérez García-Pando, C., Perlwitz, J. P. and Tegen, I.: Impact of Dust Radiative
Forcing upon Climate, in Mineral Dust, edited by P. Knippertz and J.-B. W. Stuut, pp. 327–357, Springer Netherlands, Dordrecht, https://doi.org/10.1007/978-94-017-8978-3_13, , 2014b.

Moosmüller, H., Engelbrecht, J. P., Skiba, M., Frey, G., and Chakrabarty, R. K., and Arnott, W. P.: Single scattering albedo of fine mineral dust aerosols controlled by iron concentration, J Geophys Res, 117(D11210) https://doi.org/10.1029/2011JD016909, 2012.

Morcrette, J.-J. and Fouquart, .: The overlapping of cloud layers in shortwave radiation parameterizations, J Atmos Sci, 43, 321–328, 1986.

Myhre, G., D. Shindell, F.-M. Bréon, W. Collins, J. Fuglestvedt, J. Huang, D. Koch, J.-F. Lamarque, D. Lee, B. Mendoza, T. Nakajima, A. Robock, G. Stephens, T. Takemura and H. Zhang,: Anthropogenic and Natural Radiative Forcing. Climate Change: The Physical Science Basis. Contribution of Working Group I to the Fifth
Assessment Report of the Intergovernmental Panel on Climate Change [Stocker, T.F., D. Qin, G.-K. Plattner, M. Tignor, S.K. Allen, J. Boschung, A. Nauels, Y. Xia, V. Bex and P.M. Midgley (eds.)]., University Press, Cambridge, United Kingdom and New York, NY, USA., 2013.



Nenes, A., Murray, B. and Bougiatioti, A.: Mineral Dust and its Microphysical Interactions with Clouds, in Mineral Dust, edited by P. Knippertz and J.-B. W. Stuut, pp. 287–325, Springer Netherlands, Dordrecht,
https://doi.org/10.1007/978-94-017-8978-3_12, , 2014.

Nickovic, S., Vukovic, A., Vujadinovic, M., Djurdjevic, V. and Pejanovic, G.: Technical Note: High-resolution mineralogical database of dust-productive soils for atmospheric dust modeling, Atmospheric Chem. Phys., 12(2), 845–855, https://doi.org/10.5194/acp-12-845-2012, 2012.

Pérez, C., Nickovic, S., Pejanovic, G., Baldasano, J. M. and Özsoy, E.: Interactive dust-radiation modeling: A
step to improve weather forecasts, J. Geophys. Res., 111(D16), D16206, https://doi.org/10.1029/2005JD006717, 2006.

Perlwitz, J. P., Pérez García-Pando, C. and Miller, R. L.: Predicting the mineral composition of dust aerosols – Part 1: Representing key processes, Atmospheric Chem. Phys., 15(20), 11593–11627, https://doi.org/10.5194/acp-15-11593-2015, 2015.

Reid, J. S.: Comparison of size and morphological measurements of coarse mode dust particles from Africa, J. Geophys. Res., 108(D19), 8593, https://doi.org/10.1029/2002JD002485, 2003.

Ridley, D. A., Heald, C. L., Kok, J. F. and Zhao, C.: An observationally constrained estimate of global dust aerosol optical depth, Atmos Chem Phys, 16, 15097–15117, https://doi.org/10.5194/acp-16-15097-2016, 2016.

Roehrig, R., Bouniol, D., Guichard, F., Hourdin, F. and Redelsperger, J.-L.: The Present and Future of the West
African Monsoon: A Process-Oriented Assessment of CMIP5 Simulations along the AMMA Transect, J. Clim., 26(17), 6471–6505, https://doi.org/10.1175/JCLI-D-12-00505.1, 2013.

Ryder, C. L., Marenco, F., Brooke, J. K., Estelles, V., Cotton, R., Formenti, P., McQuaid, J. B., Price, H. C., Liu, D., Ausset, P., Rosenberg, P. D., Taylor, J. W., Choularton, T., Bower, K., Coe, H., Gallagher, M., Crosier, J., Lloyd, G., Highwood, E. J. and Murray, B. J.: Coarse-mode mineral dust size distributions, composition and
optical properties from AER-D aircraft measurements over the tropical eastern Atlantic, Atmospheric Chem. Phys., 18(23), 17225–17257, https://doi.org/10.5194/acp-18-17225-2018, 2018.

Ryder, C. L., Highwood, E. J., Rosenberg, P. D., Trembath, J., Brooke, J. K., Bart, M., Dean, A., Crosier, J., Dorsey, J., Brindley, H., Banks, J., Marsham, J. H., McQuaid, J. B., Sodemann, H., and Washington, R.: Optical properties of Saharan dust aerosol and contribution from the coarse mode as measured during the Fennec 2011
aircraft campaign, 13, 303–325, 3, 2013., Atmos Chem Phys, 13(3), 303–325, https://doi.org/10.5194, 2013.

Schulz, M., Balkanski, Y. J., Guelle, W. and Dulac, F.: Role of aerosol size distribution and source location in a three dimensional simulation of a Saharan dust episode tested against satellite-derived optical thicknes, J Geophys Res Atm, 103(D9), 10579–10592, 1998.

Sheen, K. L., Smith, D. M., Dunstone, N. J., Eade, R., Rowell, D. P. and Vellinga, M.: Skilful prediction of
Sahel summer rainfall on inter-annual and multi-year timescales, Nat. Commun., 8(14996), https://doi.org/10.1038/ncomms14966, 2017.



Sokolik, I. N. and Toon, O. B.: Direct radiative forcing by anthropogenic airborne mineral aerosols, Nature, 381(6584), 681–683, https://doi.org/10.1038/381681a0, 1996.

Solmon, F., Mallet, M., Elguindi, N., Giorgi, F., Zakey, A. and Konaré, A.: Dust aerosol impact on regional precipitation over western Africa, mechanisms and sensitivity to absorption properties, Geophys. Res. Lett., 35(24), L24705, https://doi.org/10.1029/2008GL035900, 2008.

Trenberth, K. E.: Atmospheric moisture recycling: Role of advection and local evaporation, J Clim, 12, 1368–13!1, https://doi.org/10.1175/1520-0442(1999)012<1368:AMRROA, 1999.

Trenberth, K. E. and Shea, D. J.: Atlantic hurricanes and natural variability in 2005, Geophys. Res. Lett., 33(12),
L12704, https://doi.org/10.1029/2006GL026894, 2006.

Woodward, S.: Modeling the atmospheric life cycle and radiative impact of mineral dust in the Hadley Centre climate model, J Geophys Res, 106, 18155–18166, 2001.

Yoshioka, M., Mahowald, N. M., Conley, A. J., Collins, W. D., Fillmore, D. W., Zender, C. S. and Coleman, D. B.: Impact of Desert Dust Radiative Forcing on Sahel Precipitation: Relative Importance of Dust Compared to
Sea Surface Temperature Variations, Vegetation Changes, and Greenhouse Gas Warming, J. Clim., 20(8), 1445–1467, https://doi.org/10.1175/JCLI4056.1, 2007.

Yu, Y., Notaro, M., Wang, F., Mao, J., Shi, X. and Wei, Y.: Observed positive vegetation-rainfall feedbacks in the Sahel dominated by a moisture recycling mechanism, Nat. Commun., 8(1), 1873, https://doi.org/10.1038/s41467-017-02021-1, 2017.


## Author contributions

Y.B. conceived and designed the project and led manuscript writing. Y.B. and O.B. analysed the data and interpreted results. R.C.-G processed the observational satellite dataset and compared with the model. Y.B. led
manuscript writing and preparation. R.B., O.B., R.C.-G. and J.S. assisted in manuscript writing and preparation.

## Competing interests

The authors declare no competing interests.

## Availability of materials and data

The datasets generated during and/or analysed during the current study are available from the corresponding
author on reasonable request.



**Acknowledgements**

This work has received funding from the European Union's Horizon 2020 research and innovation programme under grant agreement No 641816 (CRESCENDO) to Y.B., R. C.-G. and O.B.. Y.B. and R. C.-G. acknowledge the hospitality of the Institut Pascal, UPSaclay, during the INDICES programme 2019. The simulations were performed using supercomputing resources from the GENCI (Grand Equipement National de Calcul Intensif) under grant 2020-t2014012201.


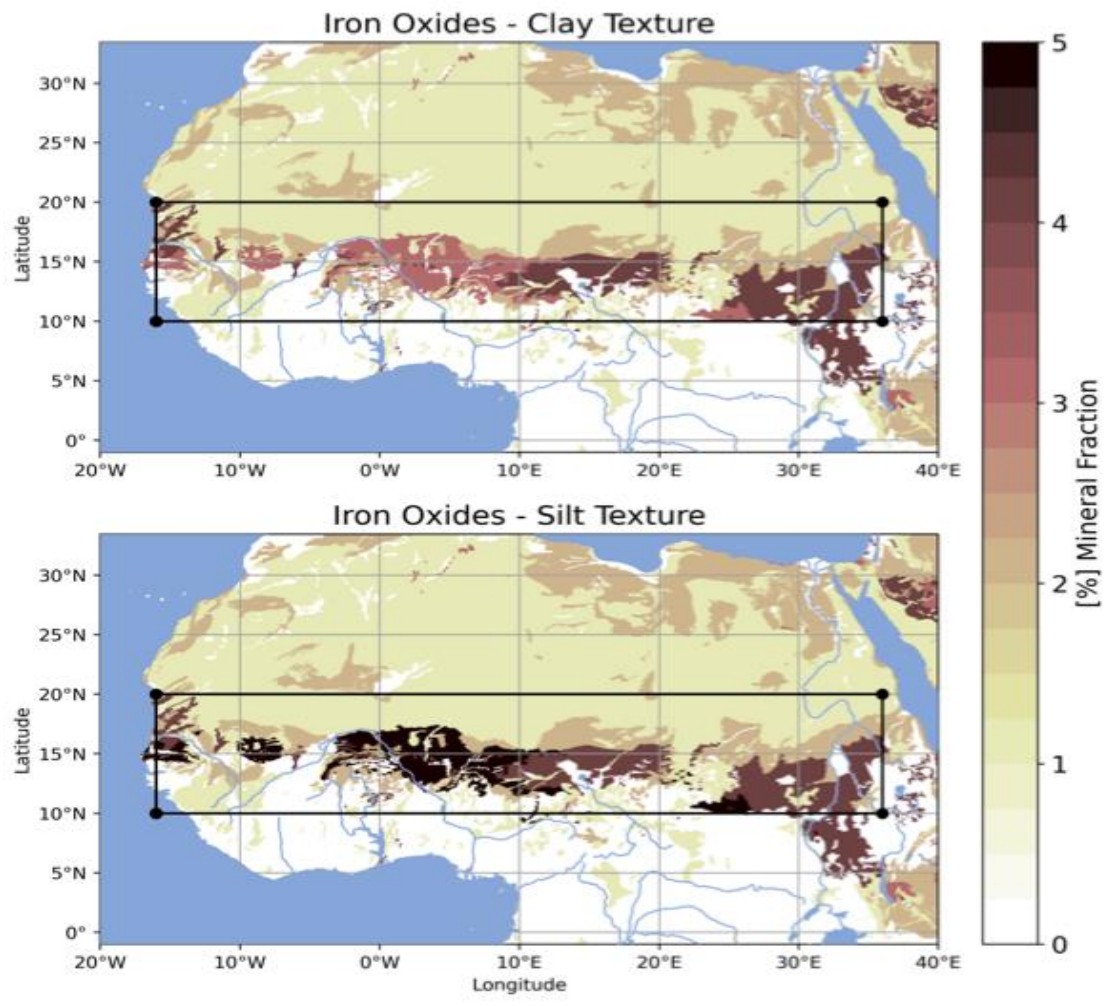

**Figure 1. Iron oxide content (by mass) in the clay and silt fractions of soils over Africa from Nickovic et al. (2012).**






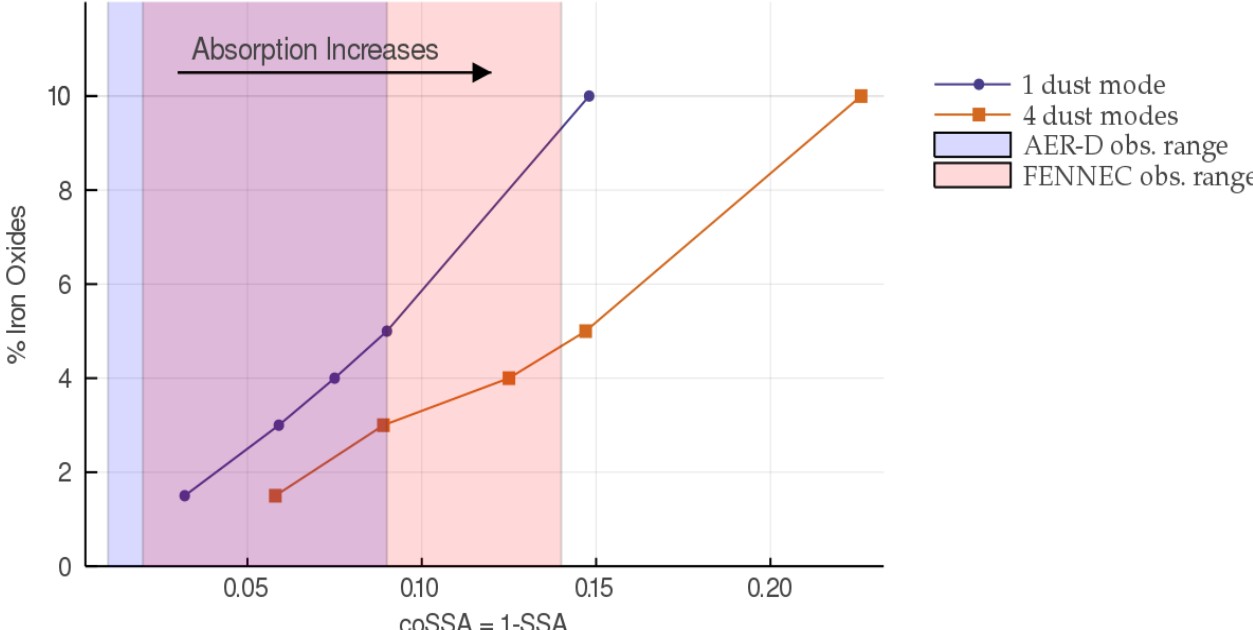

**Figure 2. Relationship between the aerosol absorption, (co-single-scattering albedo) presented on the *x*-axis, and the percentage in volume of iron oxide present in dust. The solid blue line presents this relationship for dust particles with diameters of less than 10 μm. The shift from the blue to the orange line shows the increase in dust absorption when the full dust size distribution observed in Ryder et al. (Ryder et al., 2018) is considered. The optical model used is described in the Methods. The ranges of absorptions measured during the AER-D campaign (Ryder et al., 2018) over Sahara and the near Atlantic are indicated by the blue shading (from 0.01 to 0.09 for the co-albedo), the range measured during the FENNEC campaign (Ryder et al., 2013) with the red shading (0.02 to 0.14 for the coSSA).**




**Figure 3. Top panel: Top-of-atmosphere Dust Direct Radiative Effect (in W m⁻²); Middle panel: Dust Atmospheric Absorption (W m⁻²) which is obtained as the difference between top-of-atmosphere and surface effects; Bottom Panel:**

**Surface Dust Radiative effect (W m⁻²) . The effects indicated to the left of the Figures are the sum of SW+LW, the SW and the LW for the period JJAS, respectively, Over the Sahel region (10°N to 20°N; 15°W to 35°E), at the top-of atmosphere this effect amounts to +8.1 W.m⁻² (SW=+4.4, LW=+3.7); the atmospheric absorption amounts to +25.8 W.m⁻² (SW=+36.0, LW=-10.2); at the surface, -17.7 W.m⁻² (SW=-31.6, LW=+13.9).**



**Figure 4. Mean Precipitation difference (mm d⁻¹) due to the effect of absorbing dust obtained for the months JJAS for 30 years (1985 to 2014) by substracting the precipitation fields of the experiment with dust and the experiments without dust in the IPSL-CM6 climate model where dust is run interactively.**



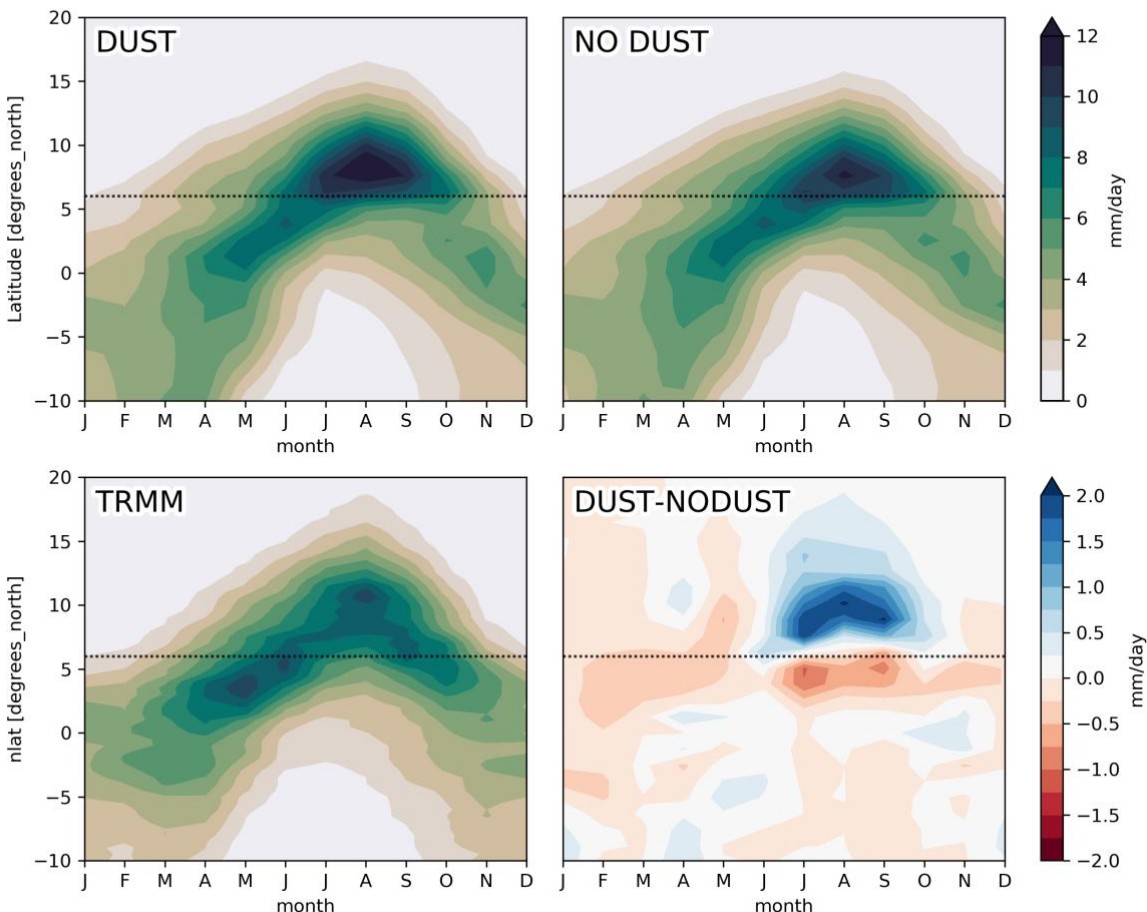

**Figure 5. Hovmoller diagram showing the time and latitudinal variations of the zonally-averaged precipitation (mm day⁻¹) from 10°W to 10°E for *JJAS* from 2000 to 2014. TRMM indicates the observed precipitations from the NASA *Tropical Rainfall Measuring Mission*.**



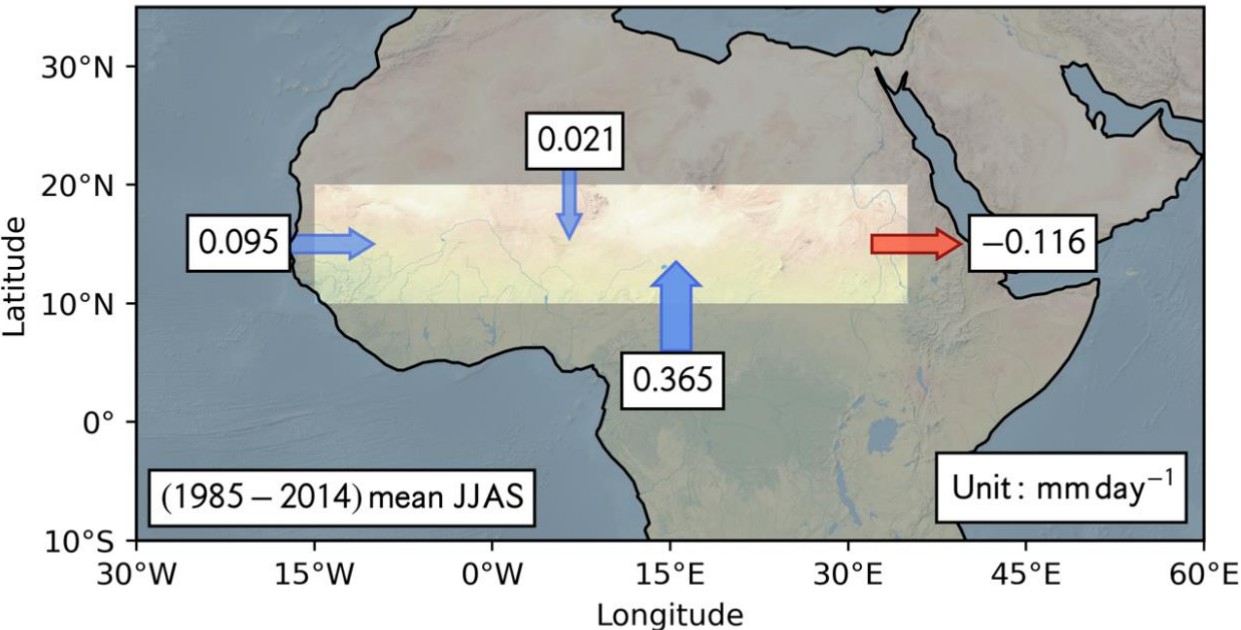

**Figure 6. Water Budget Difference (mm day[-1]) over Sahel 30-year mean (1985-2014) JJAS between with and without dust from the surface to 200 mb. The way this budget was established is explained in the Methods. The Sahel region is indicated on the Figure with a box. The difference has a positive (resp. negative) sign when water enters (resp. exits) the Sahel box.**






| Regions | No Dust vs. GPCP | | | Dust vs. GPCP Precipitation | | | Change Due to Dust |
|---|---|---|---|---|---|---|---|
| | Bias | RMSE | Correlation | Bias | RMSE | Correlation | |
| Globe | 0.277 | 1.61 | 0.821 | 0.276 | 1.62 | 0.819 | -0.1% |
| N. Atlantic (50°W–20°W; 0–30°N) | 0.625 | 1.43 | 0.952 | **0.499** | 1.25 | 0.956 | -3.9% |
| N. Africa (18W–40E; 0–35N) | 0.029 | 1.67 | 0.883 | **0.235** | 1.56 | *0.916* | 7.5% |
| Sahel (16W-36E; 10N-20N) | -1.18 | 1.51 | 0.951 | **-0.775** | **1.07** | *0.965* | 20.9% |
| West Indian Ocean (50E-70E; 10S-15N) | 1.33 | 1.74 | 0.815 | *1.26* | *1.58* | *0.865* | -2.1% |
| Equatorial Pacific (120E-90W; 10S-10N) | 0.313 | 3.67 | 0.704 | *0.326* | 3.68 | 0.709 | 0.1% |

**Table 1. Statistics of the simulated precipitation between without dust and with dust compared to the GPCP observations for the months of June-July-August-September (JJAS). Both simulated and observed precipitation fields are compared for the same JJAS period from 1985 to 2014. The Table cells with statistics in italics (resp. in bold) indicate an improvement/degradation from 5 to 15% (resp. > 15%) of the bias, RMSE, and correlation for the region indicated in the first column.**