# Peer review of "Better representation of dust can improve climate models with a too weak African monsoon"

_Atmospheric Chemistry and Physics, 2021_

## Referee Comment (RC1)

**Review of Dust Induced Atmospheric Absorption Improves Tropical Precipitations In Climate Models, by Balkanski et al, ACPD.**

**General:**

This paper brings together the idea that dust absorption is larger than previously thought owing to the presence of iron oxides and the presence of larger particles, which leads to a greater degree of solar and terrestrial absorption, heating the Sahelian region. Heating the northern hemisphere relative to the southern hemisphere (through whatever mechanism) has long been known to alter the cross equatorial energy and moisture flows leading to an increase in moisture available to the monsoon system and a northward progression of the ITCZ and vice versa (e.g. Oman et al., 2006 and Haywood et al., 2013). Putting the two things together is therefore logical, but the authors have to be careful not to overstate their results given the results come from a single model. For example the study of CMIP5 models by Huang and Frierson (2013; Figure 3) shows that not all models suffer from an ITCZ that is too far south and hence a lack of Sahelian precipitation – some have the opposite bias.

Although I rather like the idea of the paper, ultimately I was frustrated by it. It comes across as rather incomplete, not logically organised, not formatted for ACP and consequently quite difficult to decipher. It does not put the results into a wider context in terms of analysing the changes in the equatorwards transport of energy and the change in the cross-equatorial energy and moisture transport. Without this link to the more detailed physical mechanisms that have been studied by many dynamicists (e.g. papers by Kang, Frierson, Held, Huang, Hawcroft, Voigt, Schneider etc) the paper will not have the impact that its results deserve.

I conclude that, despite there being a very interesting result in the paper, the presentation is not of a suitable standard yet for publication. However, I do believe that the results are interesting and the authors should be encouraged to spend some time revising the paper as there is a good paper in there just waiting to get out…..

**Major Comments:**

Some parts of the paper (for example the refractive index and SW impacts of dust where the lead author is most familiar with the literature) are very well referenced, but other aspects are not for example the fundamentals of the ITCZ position, moisture flux diagnostics etc – more effort is required in these areas.

I would question the logic of including description of the model simulations as an Appendix. This really should be included in the body of the text. I found myself wondering how the simulations were performed, what differences there were in the simulations compared to previous simulations etc. It seems to me that the paper was possibly designed for a high impact journal, where methods are typically shunted to the end of the paper, to concentrate on the results. This is however inappropriate for ACP. The description of the modelling efforts are quite jumbled and not clear.

The SW and LW impacts are, for me, very difficult to interpret as they are not presented in a logical way. What are the SW and LW impacts at the surface and the TOA? They really should be documented better – a Table perhaps?

There are several omissions that compound the lack of completeness for example, what wavelength are you considering in Figure 2? This really does need to be stated as I can't find it in the text. This should be included both in the text and in the caption.

I completely appreciate that it must be difficult to write in a non-native language, but some of the authors (e.g. Olivier Boucher) are bilingual and would be able to sort out some of the problems with the lay-out that are currently hampering the reader's understanding.

**Specifics comments (Major and some more minor):**

Introduction: The discussion needs to include some of the more fundamental aspects of the control of the ITCZ position (see general comments). I was quite surprised to see this absent. Huang and Frierson (2013) provide one of the most accessible analyses and physical explanation of the processes that drive the Hadley circulation and the relationship between energy and moisture transport in the upper and lower branches of the Hadley cell.

L33-34: "We show here how a better representation of dust aerosols leads to an unequivocal improvement in the simulation of precipitation over key climatic tropical region" – you should state that this is for a single model.

L40: The sentence : "Conversely, (Haywood et al., 2016) discuss how some tropical precipitation biases can be reduced by changing the model's energy balance between the Northern and the Southern Hemispheres, but they did so through ad hoc hemispheric albedo changes." This makes it sounds as though there was little rationale behind the Haywood et al (2016) study. There was; the hemispheric albedo was changed so that the NH albedo = SH albedo, in agreement with hemispheric albedo data from e.g. CERES. I suggest changing the sentence to "Conversely, (Haywood et al., 2016) discuss how, in the UK Met Office HadGEM2 model, tropical precipitation biases can be reduced by setting the northern and southern hemispheric albedos to be equal in agreement with multi-decadal satellite observations."

Section 2 in general. I wasn't sure what the model simulations were being compared against. Some of this is because of the strange lay-out without a methods section (which has been demoted to an appendix). Is it 3% versus 1.5% haematite? I suspect so, but this needs to be made clearer. I wasn't sure whether the large particles had changed? This from the appendix - "Accounting for large particles of more than 10.0 μm follows a treatment of the size distribution with four modes (Di Biagio et al., 2020). The four-mode distribution has mass median diameters of 205 1.0, 2.5, 7.0 and 22.0 μm, respectively. The mineral composition which is described below is chosen to have the same dust absorption on all simulations." Do you mean that the difference between the models is that you have changed from a single mode to the four mode parameterisation above? This needs to be spelled out much more clearly.

L65. There is no indication of the wavelength at which the co-albedo is calculated. This is fundamentally important information and needs to be corrected.

L85. The statements "The difference between TOA and surface determines the dust JJAS mean atmospheric absorption due to dust (+26 W m-2 over the Sahel). Since dust is highly variable in time, particularly strong dust episodes are characterized by atmospheric absorption that reaches several hundred watts 90 per square meter (Pérez et al., 2006). Note that, in comparison, greenhouse gases contribute to a globally averaged radiative forcing of only 3 W m-2 (Myhre et al., 2013) relatively constant on short timescales."

a) I really do not think that this is a useful comparison! The radiative forcing from GHGs reported by IPCC is the CHANGE in the atmospheric concentrations since pre-industrial times.

b) I am left scratching my head about the role of LW heating within the study. This is mainly because I don't know whether the size distribution representation has been changed from the base case. The LW direct effect is not documented satisfactorily.

L95. "A general feature of most ESMs is to have a summer African monsoon that does not reach far enough North compared to observations". Could you reference this statement? Some models have the opposite bias from what I recall…. See Huang and Frierson (2013) plot.

L82-100. What are the relative roles of SW and LW? LW impacts of mineral dust are far from negligible on the monthly or seasonal means (e.g. Haywood et al., 2005) and need to be drawn out better.

Fig 3 caption suggests: The effects indicated to the left of the Figures are the sum of SW+LW. Where are these? They don't seem to be presented on the figures.

Figure 5. Why not concentrate on the summer months where the signal is more significant? It would be better to see this in more detail (see e.g. Haywood et al., 2016).

Figure 6. Is a box the best way to present the detailed response of the change in inflow of the moisture flux? It provides a simple diagnostic, but it doesn't show the details of where the additional moisture that drives the increased precipitation is coming from. I have my doubts that the moisture flux is coming from central Africa as indicted schematically by the 0.365 arrow. Moisture flux diagnostics are normally plotted as vectors (e.g. Figure S4, Haywood et al., 2016) and provide more detail about where the source of moisture is.

Table 1. Is the bias in mmday-1? This should be stated. How is the change in the bias calculated? The difference in the bias does not seem to relate to the change in the bias in a consistent way. I think that it should.

Finally, what controls the performance of the tropical precipitation appears to be the change in the cross-equatorial energy transport, which is intimately linked to the cross equatorial moisture transport (Huang and Frierson, 2013). You can see how much this improved from the "HIST" to the "STRAT" simulations in Haywood et al (2016) below when compared to CERES observations. This is not that straightforward to calculate, but at least should be referred to.

[Figure]

References:

Haywood, J.M, Allan, R.P., Culverwell I., Slingo, A., Milton, S., Edwards. J.M., and Clerbaux, N., Can desert dust explain the outgoing longwave radiation anomaly over the Sahara during July 2003? J. Geophys. Res., 110, D05105, doi:10.1029/2004JD005232, 2005.

Haywood, J.M., A. Jones, N. Bellouin, and D.B. Stephenson, Asymmetric forcing from stratospheric aerosols impacts Sahelian drought, Nature Climate Change, 3, 7, 660-665, doi: 10.1038/NCLIMATE1857, 2013.

Hwang, Y. T., and D. M. Frierson (2013), Link between the double-intertropical convergence zone problem and cloud biases over the Southern Ocean, Proc. Natl. Acad. Sci. U.S.A., 110(13), 4935–4940.

Oman, L., Robock, A., Stenchikov, G. L. & Thordarson, T. High-latitude eruptions cast shadow over the African monsoon and the flow of the Nile. Geophys. Res. Lett. 33, L18711 (2006).

---

## Author Comment (AC1)

Reply to Jim Haywood's review of "**Dust Induced Atmospheric Absorption Improves Tropical Precipitations In Climate Models**"

*We would like to thank Jim Haywood for his thorough review that helped us organize the results and their discussion in a more convincing way. We start by answering the general comments and then answer point by point his specific comments. To distinguish the review from the answers, the review is in plain text and the answers are in italics.*

**Review of Dust Induced Atmospheric Absorption Improves Tropical Precipitations In Climate Models, by Balkanski et al, ACPD.**
**General:**
    This paper brings together the idea that dust absorption is larger than previously thought owing to the presence of iron oxides and the presence of larger particles, which leads to a greater degree of solar and terrestrial absorption, heating the Sahelian region. Heating the northern hemisphere relative to the southern hemisphere (through whatever mechanism) has long been known to alter the cross equatorial energy and moisture flows leading to an increase in moisture available to the monsoon system and a northward progression of the ITCZ and vice versa (e.g. Oman et al., 2006 and Haywood et al., 2013). Putting the two things together is therefore logical, but the authors have to be careful not to overstate their results given the results come from a single model. For example the study of CMIP5 models by Hwang and Frierson (2013; Figure 3) shows that not all models suffer from an ITCZ that is too far south and hence a lack of Sahelian precipitation – some have the opposite bias.
Although I rather like the idea of the paper, ultimately I was frustrated by it. It comes across as rather incomplete, not logically organised, not formatted for ACP and consequently quite difficult to decipher. It does not put the results into a wider context in terms of analysing the changes in the equatorwards transport of energy and the change in the cross-equatorial energy and moisture transport. Without this link to the more detailed physical mechanisms that have been studied by many dynamicists (e.g. papers by Kang, Frierson, Held, Hwang, Hawcroft, Voigt, Schneider etc) the paper will not have the impact that its results deserve.
I conclude that, despite there being a very interesting result in the paper, the presentation is not of a suitable standard yet for publication. However, I do believe that the results are interesting and the authors should be encouraged to spend some time revising the paper as there is a good paper in there just waiting to get out…..

*Thanks for these encouraging remarks. We changed the title of the manuscript to reflect that dust absorption will not necessarily bring an improvement in all climate models and explicitly state that the improvements were discussed here apply to the IPSL-CM6 model. The title is now: "Dust Induced Atmospheric Absorption Influences Tropical Precipitations In IPSL-CM6 Climate Model".*

*We added the following paragraph to make the link with previous studies by dynamicists about the cross-equatorial transport of energy and tropical precipitation (lines 39 to 51 in the revised version):*
*"The intensity and the seasonal pattern of tropical precipitations are controlled by the northward cross-equatorial transport of energy (e.g. Hwang and Fierson, 2013). CERES observations of the Earth's energy budget indicate a net northward cross-equatorial transport of energy. The atmospheric component of this cross-equatorial transport is southward whereas the oceanic component is northward (see Fig. 4 in Stephens et al., 2016). Haywood et al. (2016) discussed how, in the UK Met Office HadGEM2 model, tropical precipitation biases can be reduced by setting the northern and southern hemispheric albedos to be equal in agreement with multi-decadal satellite observations. Hemispheric albedo changes thus strongly influence tropical precipitations. The link between hemispheric albedo, aerosol loadings and properties in general, and dust atmospheric absorption in particular remains however poorly understood. Here we discuss the role played by dust on tropical precipitation and describe an end-to-end physical mechanism that ties improvement in tropical precipitation to observational support for a higher level of dust absorption based on measurements of iron oxide in dust particles, measurements of the full dust particle size distribution and detailed climate simulations with interactive dust."*

**Major Comments:**
    Some parts of the paper (for example the refractive index and SW impacts of dust where the lead author is most familiar with the literature) are very well referenced, but other aspects are not for example the fundamentals of the ITCZ position, moisture flux diagnostics etc – more effort is required in these areas.
I would question the logic of including description of the model simulations as an Appendix. This really should be included in the body of the text. I found myself wondering how the simulations were performed, what differences there were in the simulations compared to previous simulations etc. It seems to me that the paper was possibly

designed for a high impact journal, where methods are typically shunted to the end of the paper, to concentrate on the results. This is however inappropriate for ACP. The description of the modelling efforts are quite jumbled and not clear.

*We regret to have left this section as an Appendix when it should have been part of the main text as the three Reviewers agree upon. We now moved the section describing the simulations and how the optical and physical aerosol properties are treated to Section 2 immediately after the introduction. We have also tried to sharpen the description of the model and the experiment design in this section.*

The SW and LW impacts are, for me, very difficult to interpret as they are not presented in a logical way. What are the SW and LW impacts at the surface and the TOA? They really should be documented better – a Table perhaps?

*Following the Reviewer's suggestion, we have added the following Table that details the SW and LW impacts at the surface, at TOA and in the atmosphere both for the Sahel region and globally. Since observational constraints exist on the dust aerosol optical depth, we have added these constraints to encourage all authors of future publications to submit a similar Table for dust radiative effects for both the Sahel region and the globe.*

| Region | DOD at 550nm (Dust Optical Depth) | Height | Dust Radiative Effect (Wm$^{-2}$) | | |
|---|---|---|---|---|---|
| | | | SW | LW | Net |
| Sahel (15°W-35°E; 10°N-20°N) | 0.37*, 5% iron oxide Dust diameter < 10μm | Top-of-atmosphere (TOA) | +3.00 | +0.37 | +3.37 |
| | | Atmospheric Absorption (TOA-Surface) | +19.5 | -0.65 | +18.8 |
| | | Surface | -16.5 | +1.02 | -15.5 |
| | 0.37*, 3% iron oxide Dust diameter < 100μm | TOA | +4.11 | +1.87 | +5.98 |
| | | Atm. Absorption | +19.9 | -3.21 | +16.7 |
| | | Surface | -15.8 | +5.09 | -10.8 |
| Global | 0.030**, 5% iron oxide Dust diameter < 10μm | TOA | -0.15 | +0.02 | -0.14 |
| | | Atm. Absorption | +1.07 | -0.04 | +1.02 |
| | | Surface | -1.22 | +0.06 | -1.16 |
| | 0.030**, 3% iron oxide Dust diameter < 100μm | TOA | -0.14 | +0.12 | -0.02 |
| | | Atm. Absorption | +1.28 | -0.29 | +0.98 |
| | | Surface | -1.42 | +0.41 | -1.01 |

**Table 1. Dust radiative effect in the different model simulations. The atmospheric radiative effects (corresponding to atmospheric absorption) are highlighted in blue.**
**\*Sahel Dust Optical depth at 550 nm was scaled to 0.37 to match the according to MODIS JJAS mean from 2000 to 2014 (both on-land and Deep-Blue products)**
**\*\*Global Dust Optical depth at 550 nm was scaled to 0.030 according to observational constraints described in Kok et al. (2017).**

There are several omissions that compound the lack of completeness for example, what wavelength are you considering in Figure 2? This really does need to be stated as I can't find it in the text. This should be included both in the text and in the caption.

*We now indicate both in the text and in the caption of Figure 2 that the wavelength considered is 550nm to be consistent with the measurements from Ryder et al. (2013, 2018).*

I completely appreciate that it must be difficult to write in a non-native language, but some of the authors (e.g. Olivier Boucher) are bilingual and would be able to sort out some of the problems with the lay-out that are currently hampering the reader's understanding.

*Thank you for your understanding, we have tried collectively to iron out these problems.*

**Specifics comments (Major and some more minor):**
Introduction: The discussion needs to include some of the more fundamental aspects of the control of the ITCZ position (see general comments). I was quite surprised to see this absent. Huang and Frierson (2013) provide one of the most accessible analyses and physical explanation of the processes that drive the Hadley circulation and the relationship between energy and moisture transport in the upper and lower branches of the Hadley cell.

*Done. See answer to the main comments above*

L33-34: "We show here how a better representation of dust aerosols leads to an unequivocal improvement in the simulation of precipitation over key climatic tropical region" – you should state that this is for a single model.

*Changed to (lines 16 to 17 in the revised version):*
*" We show that the improvement of the simulated precipitation, documented here for the IPSL-CM6 climate model, results from a thermodynamical and dynamical response to dust absorption, which is unrelated to natural variability."*

L40: The sentence : "Conversely, (Haywood et al., 2016) discuss how some tropical precipitation biases can be reduced by changing the model's energy balance between the Northern and the Southern Hemispheres, but they did so through ad hoc hemispheric albedo changes." This makes it sounds as though there was little rationale behind the Haywood et al (2016) study. There was; the hemispheric albedo was changed so that the NH albedo = SH albedo, in agreement with hemispheric albedo data from e.g. CERES. I suggest changing the sentence to "Conversely, (Haywood et al., 2016) discuss how, in the UK Met Office HadGEM2 model, tropical precipitation biases can be reduced by setting the northern and southern hemispheric albedos to be equal in agreement with multi-decadal satellite observations."

*Changed to (lines 47 to 50 in the revised version):*
*"Haywood et al. (2016) discussed how, in the UK Met Office HadGEM2 model, tropical precipitation biases can be reduced by setting the northern and southern hemispheric albedos to be equal in agreement with multi-decadal satellite observations."*

Section 2 in general. I wasn't sure what the model simulations were being compared against. Some of this is because of the strange lay-out without a methods section (which has been demoted to an appendix). Is it 3% versus 1.5% haematite? I suspect so, but this needs to be made clearer. I wasn't sure whether the large particles had changed? This from the appendix - "Accounting for large particles of more than 10.0 μm follows a treatment of the size distribution with four modes (Di Biagio et al., 2020). The four-mode distribution has mass median diameters of 205 1.0, 2.5, 7.0 and 22.0 μm, respectively. The mineral composition which is described below is chosen to have the same dust absorption on all simulations." Do you mean that the difference between the models is that you have changed from a single mode to the four mode parameterisation above? This needs to be spelled out much more clearly.

*We compare the simulation with dust and 3% haematite with the simulation with no dust. This is now more explicitly stated in the revised manuscript.*

L65. There is no indication of the wavelength at which the co-albedo is calculated. This is fundamentally important information and needs to be corrected.

*Thank you for pointing this out. We have now added in both lines 217-220 of the revised text and the caption of Figure 2 that the co-albedo is calculated at 550nm, for the same wavelength at which measurement by Ryder et al. (2013, 2018) were made.*

"*Figure 2 illustrates the influence of the iron oxide content and the size of a particle on its radiative absorption. In this Figure, the aerosol absorption increases along the x-axis with the aerosol co-single-scattering-albedo (coSSA) calculated at 550 nm.*"

L85. The statements "The difference between TOA and surface determines the dust JJAS mean atmospheric absorption due to dust (+26 W m-2 over the Sahel). Since dust is highly variable in time, particularly strong dust episodes are characterized by atmospheric absorption that reaches several hundred watts 90 per square meter (Pérez et al., 2006). Note that, in comparison, greenhouse gases contribute to a globally averaged radiative forcing of only 3 W m-2 (Myhre et al., 2013) relatively constant on short timescales."
a) I really do not think that this is a useful comparison! The radiative forcing from GHGs reported by IPCC is the CHANGE in the atmospheric concentrations since pre-industrial times.

*Agreed, this sentence has been deleted.*

b) I am left scratching my head about the role of LW heating within the study. This is mainly because I don't know whether the size distribution representation has been changed from the base case. The LW direct effect is not documented satisfactorily.
*I will add a figure that separates the LW effect from the SW effect in the supplement of the paper. Discuss also the Table that separates the LW from the SW effects.*

L95. "A general feature of most ESMs is to have a summer African monsoon that does not reach far enough North compared to observations". Could you reference this statement? Some models have the opposite bias from what I recall…. See Huang and Frierson (2013) plot.

*This statement was made from looking at Fig. 12 from Roehrig et al (2012). We have reproduced their Figure below so that the Reviewer does not have to search for this reference. Although several models show this bias, we agree with the Reviewer that the opposite bias also exist in some models. Hence, we changed the title of the paper and indicated much more clearly that the improvements in tropical precipitation may only apply to models for which the summer African monsoon does not reach far enough North compared to observations. This conclusion is conditional on a good representation of dust absorption. We hope to carry out the message that it is necessary to well describe dust optical and physical properties to capture well the African monsoon for the good reasons.*

L82-100. What are the relative roles of SW and LW? LW impacts of mineral dust are far from negligible on the monthly or seasonal means (e.g. Haywood et al., 2005) and need to be drawn out better.

*The new Table 1 explicitly calculates these effects for comparison. The relative values of the SW and LW effects are also reported in the caption of Figure 3.*

Fig 3 caption suggests: The effects indicated to the left of the Figures are the sum of SW+LW. Where are these? They don't seem to be presented on the figures.

*These effects were described in the Figure caption. We now also give the global values of dust direct radiative effect in Table 1. The caption of Figure 3 now reads:*
"*Over the Sahel region (10°N to 20°N; 15°W to 35°E), the net effect at the top-of atmosphere amounts to +6.0 W.m$^{-2}$ (SW=+4.1, LW=+1.9); the atmospheric absorption amounts to +16.7 W.m$^{-2}$ (SW=+19.9, LW=-3.2); the surface effect is -10.8 W.m$^{-2}$ (SW=-15.8, LW=+5.1). Table 1 also indicates the values for the dust global direct radiative effect.*"

Figure 5. Why not concentrate on the summer months where the signal is more significant? It would be better to see this in more detail (see e.g. Haywood et al., 2016).

*Great suggestion. We now concentrate on the summer months on the x-axis of Figure 5 as in Haywood et al., (2016).*

Figure 6. Is a box the best way to present the detailed response of the change in inflow of the moisture flux? It provides a simple diagnostic, but it doesn't show the details of where the additional moisture that drives the

increased precipitation is coming from. I have my doubts that the moisture flux is coming from central Africa as indicted schematically by the 0.365 arrow. Moisture flux diagnostics are normally plotted as vectors (e.g. Figure S4, Haywood et al., 2016) and provide more detail about where the source of moisture is.

*We agree with the Reviewer that merging the two pieces of information in a single Figure is much better. We produced such Figure which is Figure 4 in the revised manuscript.*

Table 1. Is the bias in mmday-1? This should be stated. How is the change in the bias calculated? The difference in the bias does not seem to relate to the change in the bias in a consistent way. I think that it should.

*We have added the units in the Table as well as the following sentence to the Table caption: "The last column indicates the precipitation change (mm day$^{-1}$) over the period JJAS period."*

Finally, what controls the performance of the tropical precipitation appears to be the change in the cross-equatorial energy transport, which is intimately linked to the cross equatorial moisture transport (Huang and Frierson, 2013). You can see how much this improved from the "HIST" to the "STRAT" simulations in Haywood et al (2016) below when compared to CERES observations. This is not that straightforward to calculate, but at least should be referred to.

*We show below the graph with this computation and the case with Dust and without Dust. This graph has been added to the Supplement Information of the revised article.*

References:

Roehrig, R., Bouniol, D., Guichard, F., Hourdin, F. and Redelsperger, J.-L.: The Present and Future of the West African Monsoon: A Process-Oriented Assessment of CMIP5 Simulations along the AMMA Transect, J. Clim., 26(17), 6471‒6505, https://doi.org/10.1175/JCLI-D-12-00505.1, 2013.

[Figure]

FIG. 12. Annual cycle of precipitation (mm day$^{-1}$) averaged over 10°W–10°E. A 10-day running mean was used on each dataset. Models are organized (top left)–(bottom right) from the warmest one over the Sahara (20°–30°N, 10°W–10°E) to the coldest one.

Meridional transport of moisture (ms$^{-1}$ g kg$^{-1}$) to compare with Fig. 3 from Haywood et al 2016.
The values at the Equator are respectively: 1.128 ms$^{-1}$ g kg$^{-1}$ for the run NO DUST (solid black line) and 1. 326 ms$^{-1}$ g kg$^{-1}$ for the run with DUST, hence the enhancement factor as defined by Haywood is 1.18.

FERRET (optimized) Ver.7.2
NOAA/PMEL TMAP
15-FEB-2021 17:15:05

LONGITUDE : 178.8E(−181.3) to 178.8E
TIME : 01−JAN−1985 00:00 to 31−DEC−2014 18:00

[Figure]

LATITUDE

Atmospheric Meridional Moisture Transport (ms−1 gkg−1)

---

## Author Comment (AC2)

Reply to Comment RC2 from Johannes Quaas

*We would like to thank Johannes Quaas for his thoughtful review that helped us think about ways to better compare dust properties with observations and strengthen the evaluation part of the manuscript. We start by answering his general comments and then answer point by point his specific comments point by point. To distinguish the review from the answers, we use format in plain text for the review and italics for the answers or any text from the authors.*

Balkanski et al. present a modelling sensitivity study focusing on the North African region. They implement a revised representation of dust radiative effects into an Earth system model with interactive aerosols. Specifically, revisions to the dust iron oxide content, based on observations, as well as to the dust particle size distribution, are implemented. As the authors note at some part of the manuscript, they found some interesting improvement in the precipitation climatology in the region almost accidentally, and report about these in the present manuscript.
The study is of interest to the readership of Atmos. Chem. Phys. and largely well-written (although some copy-editing will help the reading).

I have two major comments and a few specific remarks:
- Energetic analysis: I think that even if some rather elaborate analysis is presented, it would be useful to present the energetics in a more straightforward way to allow for a more stringent understanding of the mechanisms. Some of the analysis is presented in units of $W\ m^{-2}$, other figures in $mm\ day^{-1}$. I recommend to convert all figures to one of the two units. It is then interesting to note that the extra dust absorption warms the atmosphere above the Sahel by 20 W m-2, and the extra precipitation by another 1.5 mm day-1 = 42 $W\ m^{-2}$. Judging from the results that are only presented as global, annual mean numbers in Table S1, only some 20% of the solar absorption are offset by additional terrestrial cooling. Is it thus a strong reduction of the sensible heat flux, due to the surface cooling, that balances the atmospheric energy budget? Or is it indeed lateral advection of MSE out of the domain?

*Although we understand the desire to simplify units when they are equivalent, we chose to keep units of units of $W\ m^{-2}$ when presenting dust direct radiative effect. Presenting a change in precipitation in $W\ m^{-2}$ would surprise the reader. Hence we chose to keep the figures related energy fluxes in $W\ m^{-2}$ but we added a double scale for the plots of precipitation change to show the correspondence in terms of latent heat. The conversion factor from $W\ m^{-2}$ to $mm\ day^{-1}$ is 86400/2500e3.*

*The sensible heat over Sahel is reduced from $-61\ W\ m^{-2}$ to $-52\ W\ m^{-2}$.*

- Corroboration of the results: So far, the study is a bit weak in evaluation. The sensitivity study is nicely motivated by improved observations of dust mineralogical composition and size distributions, but then only one outcome is evaluated with observations, which is the precipitation. The process analysis only relies on the model itself. In my opinion, the study would be much more robust if it was possible to corroborate improvement of the model outcome also by other observations. The strong reduction in surface radiation, would it not be possible to detect and attribute it in surface radiation measurements? And/or a possibly strong impact on free-tropospheric temperatures, is there hope to find a positive impact e.g. in comparison to reanalysis? In case a strong extra LW cooling due to dust is modelled, is it maybe possible to evaluate this with, e.g., CERES retrievals?

*We extended the comparison of the model output to observation by comparing both terms of LW and SW top-of-atmosphere (toa) outgoing fluxes to CERES observations for all regions. We present the comparison below. Both the LW and SW toa outgoing fluxes are improved significantly over the Sahel, North Atlantic and North Africa. More surprisingly, these fluxes are also improved over the West Indian Ocean. We thank Johannes Quaas for pushing us to look at this comparison.*

| Regions | rlut vs CERES No Dust vs. GPCP | | | rlut vs CERES Dust vs. GPCP | | |
|---|---|---|---|---|---|---|
| | Bias | RMSE | Correlation | Bias | RMSE | Correlation |
| Globe | -3.71 | 11.0 | 0.956 | -3.65 | 10.6 | 0.959 |
| N. Atlantic (50°W-20°W; 0-30°N) | -8.17 | 10.2 | 0.915 | -7.48 | 9.12 | 0.939 |
| N. Africa (18W-40E; 0-35N) | 6.55 | 18.1 | 0.877 | 2.64 | 13.4 | 0.922 |
| Sahel (16W-36E; 10N-20N) | 26.1 | 28.0 | 0.92 | 15.7 | 17.7 | 0.947 |
| West Indian Ocean (50E-70E; 10S-15N) | -6.24 | 10.8 | 0.889 | -5.34 | 8.94 | 0.918 |
| Equatorial Pacific (120E-90W; 10S-10N) | 0.263 | 18.4 | 0.767 | 0.202 | 18.0 | 0.779 |
| Western Europe (0-50E; 35N-60N) | 1.01 | 6.95 | 0.94 | 2.12 | 6.79 | 0.95 |

| Regions | rsut vs CERES No Dust vs. GPCP | | | rsut vs CERES Dust vs. GPCP | | |
|---|---|---|---|---|---|---|
| | Bias | RMSE | Correlation | Bias | RMSE | Correlation |
| Globe | 4.0 | 16.3 | 0.917 | 3.78 | 16.0 | 0.918 |
| N. Atlantic (50°W-20°W; 0-30°N) | 10.4 | 14.1 | 0.912 | 13.7 | 16.3 | 0.932 |
| N. Africa (18W-40E; 0-35N) | -3.42 | 18.7 | 0.78 | -2.11 | 17.1 | 0.797 |
| Sahel (16W-36E; 10N-20N) | -15.1 | 19.8 | 0.801 | -13.5 | 16.0 | 0.816 |
| West Indian Ocean (50E-70E; 10S-15N) | 5.56 | 11.6 | 0.778 | 5.89 | 10.8 | 0.801 |
| Equatorial Pacific (120E-90W; 10S-10N) | 0.712 | 20.9 | 0.774 | 0.674 | 20.8 | 0.777 |
| Western Europe (0-50E; 35N-60N) | -11.7 | 15.1 | 0.932 | -11.6 | 14.9 | 0.938 |

For 2-D maps see: https://vesg.ipsl.upmc.fr/thredds/fileServer/IPSLFS/ybalkanski/C-ESM-EP/Yves_first_comparison_ybalkanski/DustPrecipitationMaps_vs_obs/atlas_DustPrecipitationMaps_vs_obs_Yves_first_comparison.html

Minor remarks:

l17 "and is unrelated" (?)

*We have modified the sentence from:*
*"We show that the improvement results from a thermodynamical and dynamical response to dust absorption is unrelated to natural variability."*
*To*
*"We show that the improvements documented here for the IPSL-CM6 climate model results from a thermodynamical and dynamical response to dust absorption, which is unrelated to natural variability."*

l68 At which wavelength is this SSA defined?

*We now indicate that the SSA is defined for 550nm in lines 127-128:*
*"Figure 2 illustrates how co-albedo (1.-SSA), SSA (single scattering albedo at 550 nm) varies with increasing iron oxide content and the effect of considering large particles (diameter > 10 µm)."*

L71 "less than 10 µm" in radius or diameter:

*Lines 171-176:*
*"In this Figure, the aerosol absorption increases along the x-axis with the aerosol co-single-scattering-albedo (coSSA) calculated at 550 nm. The coSSA is defined as:*

$$coSSA = 1.-SSA \hspace{6cm} (1)$$

*the solid blue line illustrates the absorption calculated when only dust particles with diameter less than 10 µm are considered."*

l72 What size distribution is assumed for this result? Is it an average over the simulated one? Or a certain prescribed distribution?

*We added the following information in the caption of Fig. 2:*
*"The assumed size distribution comes from observations made during the Fennec campaign by Ryder et al. (2013) and is shown in Fig.1 of Di Biagio et al. (2020)."*

l142 I recommend to denote the surface solar radiation flux similarly to the terrestrial one. Also using the partial-derivative symbol is uncommon. It should be clarified that this form of the equation only holds in some sort of equilibrium since it does not include a surface heat storage term. As noted as major comment, I think it would be useful to show also the sensible heat flux and the impact of the dust on it as a figure.

*Following the remark from the Reviewer, we changed from the partial-derivative symbol to a change 'Δ'*
*Equation 2 is now written as:*
$$\Delta F_{surf}^{SW} + \Delta F_{surf}^{LW} + \Delta LE + \Delta S_E \approx 0$$
*To illustrate how accurate this equation was, we now added Table 3 to the manuscript.*

| Flux Differences (W m⁻²) between the cases 'Dust 3% Haematite' and 'No Dust' | | | | |
|---|---|---|---|---|
| | $\Delta F_{surf}^{SW}$ | $\Delta F_{surf}^{LW}$ | $\Delta LE$ | $\Delta S_E$ | $\Delta F_{surf}^{SW} + \Delta F_{surf}^{LW} + \Delta LE + \Delta S_E$ |
| Annual Global | -1.57 | 0.74 | 0.25 | 0.61 | -0.03 |
| Annual Sahel | -26.85 | 18.48 | -2.87 | 11.10 | -0.14 |
| JJAS Global | -1.34 | 0.86 | 0.09 | 0.48 | 0.08 |
| JJAS Sahel | -23.43 | 18.44 | -2.71 | 7.66 | -0.05 |

*Table 3. Annual and JJAS flux differences estimated globally and for the Sahel region (15°W-35°E; 10°N-20°N) for the net SW and LW fluxes at the surface, the latent heat and sensible heat fluxes averaged over the 30-year period from 1985 to 2014.*

L176 "too little"

*Corrected.*

l193 In the introduction, the authors explain also the importance of dust for aerosol-cloud interactions. It would be useful if this model description section clarified what aspects of such interactions are represented in the IPSL model with which parametrisations.

*We added to following sentences in lines 127-130: "In the IPSL-CM6 model, dust is considered hydrophobic although laboratory measurements have shown that dust can act as condensation nuclei in certain environmental conditions (Nenes et al., 2014). The parametrization of dust acting as an ice-seeding particle has yet to be included."*

L200 "several' modes – is this an arbitrary number?

*We now indicate lines 81 to 83 that they are 4 modes that represent the size distribution. This new size-distribution representation was published in Di Biagio et al. (2020).*
*"The size distribution is represented by one or by four modes, each one represented with a log-normal distribution consisting of a mass median diameter which varies in response to the sink processes that affect the dust cycle."*

L201 "either" not followed by an "or"?

*We modified this sentence accordingly.*

L 240 This seems to be a mistake, as Table 1 is the comparison of precipitation statistics.

*We had omitted a reference to Balkanski et al. (2007), which we have now added:*
*"We refer the reader to Table 1 of Balkanski et al. (2007) that explains the abundances of the different assemblages and minerals."*

L250 June typically is defined as part of boreal summer, but September usually isn't.

*When we reviewed the literature on the West African Monsoon, many studies covered the period JJAS instead of the boreal summer (JJA). We took out the term 'summer'.*
*Lines 134 to 136: "We analyzed the last 30 years of the coupled simulations (1985-2014) for the period when precipitation is most abundant over the Sahel from June to September referred to as JJAS in the rest of the text."*

L258 Reference is missing
*The following reference has been added:*
*Fouquart, Y. and Bonnel, B.: Computations of Solar Heating of the Earth's Atmosphere—A New Parameterization, Beiträge zur Physik Atmosphäre, 53, 35–62, 1980.*

L284 "\rho"; q and u should be explained separately, and u needs to be written as a vector.

*We added the following text, lines 161-162: " where $\rho_w$ is the density of water, $g$ is the acceleration due to gravity, $p_s$ is the surface pressure, $\langle q\vec{u} \rangle$ represents the monthly mean of the product of the water content times the wind speed."*

l292 What is the \delta for in this equation? Also in the vertical gradient, it should be a "\partial p" in the denominator.

*Following the recommendation of the Reviewer, this moist static energy change is no longer shown in this manuscript.*

L294 Why not q rather than 'ovap'?

*This variable does not appear anymore*

l382-394 Double references

*We have suppressed the double references.*

L491 It is not understandable what is meant by "The effects indicated to the left of the Figures…"

*This sentence has been deleted*

l507 The units need to be provided

*We now indicate that the precipitation changes are in mm.day⁻¹*

Fig. S1 I do not understand the colour coding. The legend says this is the transport at 800 mb over oceanic regions, yet the height is also expressed in mb. How can this be isobaric over ocean but then change pressure height as soon as it crosses the shoreline? Also it seems not overly useful to analyse (a) in terms of RH, and (b) at one level only. Why not rather corroborate the budget analysis shown in Fig. 6 by a supplementary figure that shows the vectors of the moisture flux as integral from the surface to 200 mb?

*The pressure follows terrain coordinates hence the change over land. We have merged this Figure with the one showing the changes in precipitation due to dust absorption (Figure 4) following the recommendation of Reviewer Jim Haywood.*

Fig. S2: I do not find this figure very instructive, and the authors seemingly not either: it is almost not explained and discussed in the manuscript, and accordingly I have difficulties understanding what the authors want to convey with it.

*We agree and we have removed this Figure from the paper.*

References :

Boucher, O., J. Servonnat, A. L. Albright, O. Aumont, Y. Balkanski, V. Bastrikov, S. Bekki, R. Bonnet, S. Bony, L. Bopp, P. Braconnot, P. Brockmann, P. Cadule, A. Caubel, F. Cheruy, F. Codron, A. Cozic, D. Cugnet, F. D'Andrea, P. Davini, C. de Lavergne, S. Denvil, J. Deshayes, M. Devilliers, A. Ducharne, J.-L. Dufresne, E. Dupont, C. Ethé, L. Fairhead, L. Falletti, S. Flavoni, M.-A. Foujols, S. Gardoll, G. Gastineau, J. Ghattas, J.-Y. Grandpeix, B. Guenet, L. Guez, E. Guilyardi, M. Guimberteau, D. Hauglustaine, F. Hourdin, A. Idelkadi, S. Joussaume, M. Kageyama, A. Khadre-Traoré, M. Khodri, G. Krinner, N. Lebas, G. Levavasseur, C. Lévy, L. Li, F. Lott, T. Lurton, S. Luyssaert, G. Madec, J.-B. Madeleine, F. Maignan, M. Marchand, O. Marti, L. Mellul, Y. Meurdesoif, J. Mignot, I. Musat, C. Ottlé, P. Peylin, Y. Planton, J. Polcher, C. Rio, N. Rochetin, C. Rousset, P. Sepulchre, A. Sima, D. Swingedouw, R. Thieblemont, A. Traoré, M. Vancoppenolle, J. Vial, J. Vialard, N. Viovy, and N. Vuichard, Presentation and evaluation of the IPSL-CM6A-LR climate model, *Journal of Advances in Modeling Earth System,* https://doi.org/10.1029/2019MS002010.

Di Biagio, C., Balkanski, Y., Albani, S., Boucher, O., & Formenti, P. (2020). Direct radiative effect by mineral dust aerosols constrained by new microphysical and spectral optical data. *Geophysical Research Letters, 47,* e2019GL086186. https://doi.org/ 10.1029/2019GL086186.

Ryder, C. L., Highwood, E. J., Rosenberg, P. D., Trembath, J., Brooke, J. K., & Bart, J. K. (2013). Optical properties of Saharan dust aerosol and contribution from the coarse mode as measured during the Fennec 2011 aircraft campaign. *Atmospheric Chemistry and Physics,* 13(1), 303–325. https://doi.org/10.5194/acp-13-303-2013.

---

## Author Comment (AC3)

**Answers to Comment RC3**

*We would like to thank Reviewer RC3 for his/her thoughtful review that helped us improve the manuscript and clarify the simulations presented. We start by answering the general comments and then make a point by point answer to his/her specific comments. To distinguish the review from the answers, we use format in plain text for the review and italics for the answers or any text from the authors.*

This study adds new large dust particle sizes to the IPSL-CM6 climate model simulations. Model computations that include these larger particle sizes produce more atmospheric heating and affect the transport of moisture over the Sahel; this results in improved distribution of modeled precipitation rates (in comparisons to observations). This is an interesting paper with significant potential, but there are some inconsistencies in the paper that weaken the overall message.

For instance, although Section 5.2 states that the dust simulations are either done with one mode to represent both the accumulation and coarse modes or with four modes that encompass a much larger size range, I do not see any analysis that uses the single-mode runs in the paper. Everything refers to "with dust" and "without dust." None of the figures or tables present differences associated with single- vs multi-mode dust.

*We now clearly indicate that dust simulations were performed both with a size distribution without coarse particles of more than 10 μm (1-mode run), and with a size distribution that includes the coarser particles from <1 to 100 μm range (4-mode run). These simulations were made comparable by normalizing the globally-averaged dust optical depth at 550nm to the value of 0.030 as discussed in Kok et al. (2017).*

Other issues that should be addressed (majors):

One thing that bothered me throughout the paper is that the authors assume that the wind-blown particles have hematite concentrations of about 3% by volume in both the clay and silt fractions (per lines 81 and 225), citing Nickovic (2012). (At line 231 they vary the volume fraction of hematite from 0.9 to 10%, but this sensitivity test is not mentioned anywhere else in the paper). Thus, they assume constant iron-oxide mineralogies wrt size (first presented around lines 70-80). We don't necessarily expect the silts to have the same composition as clays, though. Some discussion about the robustness of assuming that all particle sizes have the same composition would improve the value of this paper. A literature search for measurements that provide the relative abundance of hematite/goethite in clays and silts would be worthwhile. If you can find measurements that demonstrate that the proportion of iron is constant wrt size, that will make this work stronger. If it turns out that this is not the case, then an adjustment to your analysis that reflects this variability will make this work stronger. If the literature is ambiguous, then it would be useful to also include non-absorbing silt as part of the analysis (to show the minimal effect of the large particles).

*To our knowledge, they are no existing measurements of how the amount of iron oxide (in particular hematite) varies with size. Some authors have speculated (and we are amongst them) that the iron oxide content decrease from the clay to the silt fraction but no hard measurements have confirmed this yet. Rather than speculating on this, we have preferred to start from established results documented in the literature. We found that Nickovic et al. (2012) had formulated a method to compute the iron oxides separately for the estimated clay and silt fractions. We used his results and we added a sentence to clearly indicate it to the reader.*

As I read this paper, it was not clear to me whether the addition of the larger particles increased the model emissions, or if the modeled emissions were scaled to larger sizes. This should be stated very explicitly, because adding more particles to the system will of course increase absorption, regardless of particle size. More particles --> more AOD --> more AAOD --> more radiative effect, even if SSA is held constant.

*We relied on a previous publication that two of us coauthored (Di Biagio et al., 2020), where we constrained the emissions of dust according to the constrained of Kok et al. (2017) discussed above. We thank the Reviewer for pointing out that we need to be more explicit about the amount of dust emitted in both cases. We now indicate that for particles with diameters of less than 10 μm represented with a mono-modal size distribution (MMD=2.5 μm, sigma=2.0) the total yearly emissions of dust are 1,764 Gtons yr$^{-1}$ whereas for the 4-modes size distribution representing the particles up to*

100 μm these emissions amount to 18,122 Gtons yr $^{-1}$. Both simulations have the same gobally-averaged dust aerosol optical depth at 550nm of 0.030 following Kok et al. (2017).

Also, Figures 4 & 5 discuss the effect of "dust" vs "no dust," but elsewhere in the paper the importance of large particle absorption is emphasized. The thing that is missing from this paper is a comparison of "dust with large particles" vs "dust without large particles." Alternatively, I would like to see "dust with large particles" vs "no dust" AND "dust without large particles" vs "no dust." The way that the paper is written right now, though, the effect of the newly added large particles is still unquantified.

*Indeed, we had not clearly stated in the original manuscript that we had two equivalent cases in terms of radiative effect of dust at top-of-atmosphere, at the surface and in the atmosphere. We have now added a Table (Table 1, see below) and clearly state in the text the equivalence between two cases: 1-mode and 5% iron oxide content, and, 4 modes and 3% iron oxide content. We thank the Reviewer for helping clarify this key point in the study.*

Lines 111-113: I don't understand this sentence wrt Figure 6; Fig 6 show a large \*positive increase\* in water flux (+0.365) at the southern border of the Sahel. I understand that -0.41 - (-.05) = -0.36, but this is still not consistent with the +0.365 of the figure.

*The sign convention as indicated in the caption of this Figure is a positive water flux for the water entering the Sahel region and a negative water flux when water is exiting the Sahel. Hence this value of +0.365 mm day $^{-1}$ is a net flux entering the South boundary of the Sahel region. The last sentence of the caption of Figure 6 in the revised manuscript has been changed to: "The sign of the water budget difference is positive (resp. negative) if water enters (resp. exits) the Sahel box." to make this clearer:*

Line 114-115: I don't see 0.40 mm/day anywhere in Fig 6. Overall, I am having difficulty aligning the energy budget of Fig 6 with the text.

*The 0.40 mm day $^{-1}$ is not on Fig. 6 since it is the difference in precipitation between the simulation with dust and the simulation without dust. Figure 6 only presents the terms of advection of water vapor into the Sahel region, not the terms of precipitation and evaporation differences resulting from the presence of dust. We now indicate the following in lines 232 to 236: "The change in water flux into the Sahel region amounts to an increase of 0.365 mm day $^{-1}$ for the JJAS period. Precipitation and evaporation over the Sahel region increase by 0.40 and 0.09 mm day $^{-1}$, respectively, for the same months over the 30-year period (1985-2014). Hence there is a small residual in the water budget which we attribute to imperfections in the way the advected water fluxes are diagnosed in the model."*

+ On line 224 the authors state that they used 3% iron oxides by volume. Then on line 230 they state that they vary the volume of hematite from 0.9 to 10%. The remainder of the text, however, does not discuss the sensitivity of varying the hematite fraction.

*The case studied in the paper is 3% iron oxides by volume. We now state it mode clearly. We think that the variations we studied are useful to understand how we proceeded and clearly indicate that we are interested in dust from the Sahel region with a content of 3% iron oxides by volume.*

+ On line 240, the authors state "We refer the reader to Table 1 that explains the abundancies of the different assemblages and minerals.", but I do not see this information in Table 1 or in any other table.

*This was also pointed out by Johannes Quaas. We had omitted a reference to Balkanski et al. (2007), which we have now added in the revised manuscript: "We refer the reader to Table 1 of Balkanski et al. (2007) that explains the abundancies of the different assemblages and minerals. "*

+ Figures 2 and 6 refer to a "Methods" section that does not exist.

*We regret to have left this section that we referred to as "Methods" as an Appendix when it should have been part of the main text as the three reviewers agree upon. We have now moved the section describing the simulations and how the optical and physical properties are treated to Section 2 immediately after the introduction. We also tried to improve this section.*

+ Figure 3 caption mentions "The effects indicated to the left of the Figures...", but I do not see anything to the left of the figures.

*We added the following Table that details the SW and LW impacts at the surface, at TOA and the atmospheric absorption both for the Sahel region and globally. Since observational constraints exist on the dust aerosol optical depth, we added these constraints to encourage all authors of future publications to submit a similar Table for dust radiative effects for both the Sahel region and the globe.*

| Region | DOD at 550nm (Dust Optical Depth) | Height | Dust Radiative Perturbation (Wm$^{-2}$) | | |
|---|---|---|---|---|---|
| | | | SW | LW | Net |
| Sahel (15W:35E; 10N:20N) | 0.37*, 5% iron oxide Dust diameter < 10μm | Top-of-atmosphere (TOA) | +3.00 | +0.37 | +3.37 |
| | | Atmospheric Absorption (TOA-Surface) | +19.5 | -0.65 | +18.8 |
| | | Surface | -16.5 | +1.02 | -15.5 |
| | 0.37*, 3% iron oxide Dust diameter < 100μm | TOA | +4.11 | +1.87 | +5.98 |
| | | Atm. Absorption | +19.9 | -3.21 | +16.7 |
| | | Surface | -15.8 | +5.09 | -10.8 |
| Global | 0.030**, 5% iron oxide Dust diameter < 10μm | TOA | -0.15 | +0.02 | -0.14 |
| | | Atm. Absorption | +1.07 | -0.04 | +1.02 |
| | | Surface | -1.22 | +0.06 | -1.16 |
| | 0.030**, 3% iron oxide Dust diameter < 100μm | TOA | -0.14 | +0.12 | -0.02 |
| | | Atm. Absorption | +1.28 | -0.29 | +0.98 |
| | | Surface | -1.42 | +0.41 | -1.01 |

**Table 1. Dust radiative perturbation in the different model simulations. The atmospheric radiative perturbations (corresponding to atmospheric absorption) are highlighted in blue.**
**\*Sahel Dust Optical depth at 550 nm was scaled to 0.37 to match the according to MODIS JJAS mean from 2000 to 2014 (both on-land and Deep-Blue products)**
**\*\*Global Dust Optical depth at 550 nm was scaled to 0.030 according to observational constraints described in Kok et al. (2017).**

Other issues (minor):

In some ways it was nice to jump right to the results and discussion (without first presenting the methodology), but it is unusual for an ACP article. We usually see this in very compact articles that target a larger range of scientific disciplines. Such articles use this format because scientists outside of a certain specialty may have little knowledge or interest in the exact methods, but that is generally not the case with ACP audiences. Additionally, the format was problematic because the authors kept referring to a section called "Methods," but no such section exists in this article. Eventually I figured out that the "Methods" section is the Appendix.

*We regret to have left this section referred to as "Methods" in the text that is in fact the Appendix when it should have been part of the main text. We have now moved the section describing the simulations and how the optical and physical properties are treated to Section 2 immediately after the introduction.*

The readability of the article would be much better if the contents of the Appendix immediately followed the introduction, in my opinion. Readers can skip this part and return to it later, if they choose (and many will). A 2nd choice would be to put the methodology after the Conclusions but not in the Appendix. An appendix is a supplement, so to some extent it is superfluous. Methodology, on the other hand, is not superfluous. Finally, if the authors are adamant about keeping this material in the Appendix, I recommend that they make the name of the Appendix descriptive (e.g., Appendix: Methodology).

We are aware that this was not optimal as the three reviewers pointed out. We have now moved the section describing the simulations and how the optical and physical properties are treated to Section 2, immediately after the introduction. We have also included and improved the description of the simulations.

Figure S1 is a nice visual that can help readers understand material in the main text. It should be moved to the main body of the article, in my opinion.

*We merged this information into the new Figure 6 by now showing the wind vectors on top of the budget of water advected to the Sahel.*

Lines 88-91: Comparing episodic absorption of dust to global forcing of greenhouse gases is a bit of an apples-to-oranges hoodwink, eh?

*We agree and have deleted the sentence where this not-so-welcomed comparison was done.*

Lines 232-241 seems like a complicated way of using the Maxwell Garnett (MG) effective medium approximation (EMA). Bohren and Huffman (1983) present a nice equation and discussion about the MG EMA for multi-component mixtures. It is on page 216 in my paperback version (between Equations 8.49 and 8.50).

*We simplified this formulation.*

The discussion begins with "We now compare the distribution of the surface precipitation between the two model simulations... " This comes as a complete surprise, since the authors have already discussed all six figures. I thought that we had already seen comparisons between "with dust" and "without dust" simulations?

*In this part of the manuscript, we compare the precipitation fields to observations using standard statistics: bias, the root mean square error (RMSE) and the spatial correlation. We changed the sentence to: "We now seek to analyze if the changes in precipitations brought about by the presence of dust improve or degrade the satistics of surface precipitations when compared to observations. We selected the observations from the Global Precipitation Climatology Project (GPCP) from June to September over the period 1985 to 2014."*

Line 130: ..."(larger particles being more absorbing than smaller ones)." Since the authors have not demonstrated that this is the case (at least they have not provided large/small comparisons thus far), they should provide the reader with a citation to another study.

*We now cite the measurements made by Claire Ryder during the FENNEC campaign (Ryder et al., 2013) that illustrate how much more absorption is caused by large dust particles compared to smaller ones.*

There article could use further proof reading, in places.

*The authors worked collectively on the text to improve its wording and the logic of the different parts.*

+ Lines 69-81 a little bumpy.

*Lines 71 through 81 were deleted as they did not bring new information.*

+ Line 100: bumpy

*We replaced this line with the following sentences (ll 239-244 ): "We compare the SW and LW radiative effects of two simulations that have the same absorption: 4 modes (including large particles, 10um < D < 100um) with 3% iron oxides, to the simulation with 1 (without large particle, 10um < D < 100um), the results are shown in Table 1. We ran for the full 100 -year period only the simulation with 1 mode and 5% iron oxide content equivalent to the full size distribution (4 modes) and 3% iron oxide. "*
The addition of Table 1 reinforces this point.

+ Line 104-106: bumpy

*We changed this sentence to: "Hence, the substantial increase in aerosol absorption caused by large particles will be particularly marked over dust source regions."*

+ On line 232 they discuss a Maxwell-Bruggeman approximation; I suspect that they really mean the Maxwell Garnett approximation, as in Balkanski (2007).

*Thank you, this has been corrected.*

+ Lines 242-252 in the Appendix are redundant with the main text.

*We eliminated this redundancy.*

+ Line 284 has a variable missing.

*It is now corrected.*
* * *
Bohren, C., and D. Huffman (1983), Absorption and scattering of light by small particles, Wiley.
* * *
Citation: https://doi.org/10.5194/acp-2021-12-RC3

**References**

*Di Biagio, C., Balkanski, Y., Albani, S., Boucher, O. and Formenti, P.: Direct Radiative 335 Effect by Mineral Dust Aerosols Constrained by New Microphysical and Spectral Optical Data, Geophys. Res. Lett., 47(2), https://doi.org/10.1029/2019GL086186, 2020.*

*Kok, J. F., Ridley, D. A., Zhou, Q., Miller, R. L., Zhao, C., Heald, C. L., Ward, D. S., Albani, S. and Haustein, K.: Smaller desert dust cooling effect estimated from analysis of dust size and abundance, Nat. Geosci., 10(4), 274– 278, https://doi.org/10.1038/ngeo2912, 2017.*

---

## Author Response (AR2)

Answers to the Editor Ulrich Poeschl and to the Reviewer Jim Haywood

We would like to thank both the Editor and the Reviewer for helping us carry through more improvements to this manuscript. We have now considered all their suggestions and answer point by point to them. The suggestions and in italics, our answers are in blue in plain text.

Suggestions from the Editor:

*Dear Yves and Colleagues,*

*On behalf of the ACP executive committee, I will be happy to accept your manuscript for final publication as an ACP Letter if you can properly address the following comments and revise the manuscript accordingly:*

*1) It might be worth working with the authors to find a better title. The current title is "Dust Induced Atmospheric Absorption Influences Tropical Precipitations In IPSL-CM6 Climate Model". Aside from the two grammatical errors (should be Dust-induced and Precipitation), the term "atmospheric absorption" is meaningless. To people working on radiation it might be obvious that it refers to solar radiation. I also don't like the restriction to one model (requested by the referee). The authors make some general points about ESMs, so it would be good if they could define a title that makes a broader point while not over-extending to imply all models.*

Following your suggestion, we iterated between the authors and came up with a title without grammatical errors that is not restricted to one climate model and takes into consideration the point from Reviewer Jim Haywood:'' Better representation of dust can improve climate models with a too weak African monsoon''

*2) I am surprised that neither referee picked up on the terminology. The first sentence "Mineral dust influences precipitation through direct radiative forcing (Miller et al., 2014), changing the vertical temperature profile" should really read "Mineral dust influences precipitation through aerosol-radiation interaction (Miller et al., 2014), changing the vertical temperature profile".*

We corrected this sentence accordingly and looked throughout the text for possible other occurrences.

*3) Later it is stated that "absorption causes a change in atmospheric radiation of several tens of watts per square meter". Where? I presume they mean at the surface, but the surface and TOA effects strongly depend on the SSA. And why not say positive or negative? Then it says "an effect stronger than the one exerted by aerosol-cloud interactions." I presume they mean anthropogenic ACI rather than dust ACI? So are they comparing ALL dust with anthropogenic ACI? I would ask the authors to go through the paper to check for other ambiguities like this. It's unfortunate that they were not picked up in review.*

We realized from your comment that aerosol absorption needed to be defined. Once defined, it becomes clear that this quantity is always positive and hence we do not need to indicate its sign. The following sentence was added in lines 30 to 34: "The energy absorbed by mineral dust in the atmospheric column is defined as the difference between top-of-atmosphere and surface radiative effects. In source regions, it represents several tens of watts per square meter ($W.m^{-2}$) and hence is stronger than the anthropogenic aerosol-cloud interactions (Miller et al., 2014; Nenes et al., 2014). The Sahel precipitation is influenced by the energy absorbed by aerosols (Miller et al., 2004; Solmon et al., 2008; Yoshioka et al., 2007), and dust absorption of energy depends on iron oxides (hematite and goethite) that are part of dust mineralogical composition (Sokolik and Toon, 1996; Claquin et al., 1999)."

*4) In various places they talk about radiative forcing when really they mean radiative perturbation or effect. For example on line 487 they say "To evaluate the impact of this change of radiative forcing". Radiative forcing is almost always defined as pre-industrial to present day (or some decadal period), whereas I think they mean the radiative effect of dust (with/without). They need to be much more careful with their definitions/terminology.*

I apologize for leaving out these terms which are inappropriate here. They have been now changed to 'dust radiative effect'.

*5) In line 297 they state that gz is the geopotential height, but I think they mean the geopotential.*

You must have been working on an earlier version of the manuscript since the lines that erroneously stated that gz was the geopotential height did not appear anymore in the revised version of the manuscript (acp-2021-12-manuscript-version4.pdf).

*Please consider and follow up on the above comments and suggestions, revise the manuscript accordingly, and let us jointly consider better options for the title.*

Please let us know if this new title convey correctly the work described in this article.

*Many thanks, best wishes, and so long,*
*Uli*

Thank you Uli for reading this manuscript thoroughly!

Points raised by the Reviewer Jim Haywood

*This manuscript is vastly improved. I have read the response to reviewers' comments and the new manuscript file in some detail. The authors should be congratulated for making such a good job of the improvements. There are a few minor issues and a few typos that the authors should probably correct - I have included this as technical corrections.*

Guided by the comments of the three reviewers we focused on describing more accurately the simulations and on introducing early the implementation of the updated dust properties. We are happy to have convinced all three reviewers.

*L13 (and throughout the manuscript): tropical precipitations -> tropical precipitation (precipitation is a plural form of precipitation)*

Corrected

*L13 (and throughout): Sahel -> the Sahel*

This correction was made for all instances where 'Sahel' appeared in the text.

*L89: "two particle size class the clays" -> "two particle size classes: the clays"*

Corrected

*L182: The reference to Pérez et al., 2006. Agreed that there are simulated impacts on the surface SW flux of several hundred Wm-2. However, it would be useful to reference an example of a dust event that actually measured (rather than simulated) such a reduction using broadband radiometers. Note Perez (2006) explicitly acknowledge that they do not include the larger particles in their simulations "However, the study has several limitations that should be improved in the future. Currently, the size bin distribution includes dust particles smaller than 10 mm. Although the lifetime of larger particles is short, they still could significantly modify the radiative balance over emission areas. Thus it is planned to extend the range of particle toward larger particles."*

*I would suggest including more directly inferred measurements as these do not rely on accurate simulations of the dust size distribution : "Since dust is highly variable in time, this average value is consistent with values of several hundred watts per square meter simulated or inferred from surface-based measurements during particularly strong dust episodes (Pérez et al., 2006; Milton et al., 2008)."*
*Milton et al (2008) documented a strong dust event on 3rd March 2006 which reduced the downwelling surface solar broadband irradiance by >350Wm-2.*

We followed the reviewer comment and have included these sentences (lines 186-189).

Milton, S. F., G. Greed, M. E. Brooks, J. Haywood, B. Johnson, R. P. Allan, A. Slingo, and W. M. F. Grey. "Modeled and observed atmospheric radiation balance during the West African dry season: Role of mineral dust, biomass burning aerosol, and surface albedo." Journal of Geophysical Research: Atmospheres 113, no. D23 (2008).

Figure 4, is a great addition compared to the previous analysis as it shows where the moisture comes from. I am a little surprised that you chose to present RH flux (% which is not intuitive at it is temperature dependent) rather than absolute humidity (g/m3) or specific humidity (g/kg) – the units would be more intuitive.

We agree that this Figure eases understanding how the moisture is transported to the Sahel.

L269. I would say that it is a cause rather than a symptom…..

We changed the sentence (line 276) to indicate that it is a cause and not a symptom